# Exponential Objective Decrease in Convex Setup is Possible! Gradient Descent Method Variants under $(L_0, L_1)$-Smoothness

## Abstract

The gradient descent (GD) method – is a fundamental and likely the most popular optimization algorithm in machine learning (ML), with a history traced back to a paper in 1847 Cauchy (1847). It was studied under various assumptions, including so-called $(L_0, L_1)$-smoothness, which received noticeable attention in the ML community recently. In this paper, we provide a refined convergence analysis of gradient descent and its variants, assuming generalized smoothness. In particular, we show that $(L_0, L_1)$-GD has the following behavior in the *convex setup*: as long as $\left\| \nabla f(x^k) \right\| \geq \frac{L_0}{L_1}$ the algorithm shows *exponential objective decrease*, and when $\left\| \nabla f(x^k) \right\| < \frac{L_0}{L_1}$ is satisfied, $(L_0, L_1)$-GD has standard sublinear rate. Moreover, we also show that this behavior is common for its variants with different types of oracle: *Normalized Gradient Descent* as well as *Clipped Gradient Descent* (the case when the full gradient $\nabla f(x)$ is available); *Random Coordinate Descent* (when the gradient component $\nabla_i f(x)$ is available); *Random Coordinate Descent with Order Oracle* (when only $\text{sign}[f(y) - f(x)]$ is available). In addition, we also extend our analysis of $(L_0, L_1)$-GD to the strongly convex case. We explicitly confirm our theoretical results through numerical experiments.

## 1 Introduction

We consider the standard unconstrained minimization

$$\min_{x \in \mathbb{R}^d} f(x), \tag{1}$$

where $f : \mathbb{R}^d \to \mathbb{R}$ is a convex differentiable function. This problem configuration is quite general and encompasses a broad range of applications in ML scenarios. For such problems, the traditional optimization algorithm is the *gradient descent method* (GD) Cauchy (1847), which has a sublinear convergence rate in the convex setting under the Lipschitz smoothness assumption (see, e.g., Nesterov, 2013). In particular, GD is the core of optimization for machine learning, and various modifications of this method have been studied in different assumptions suited to ML applications.

In this paper, we consider one of such assumptions called $(L_0, L_1)$-*smoothness* (Zhang et al., 2020b;a; Chen et al., 2023), which in the case of twice differentiable functions, states that $\left\| \nabla^2 f(x) \right\| \leq L_0 + L_1 \| \nabla f(x) \|$, i.e., the smoothness constant can grow as a linear function of the gradient norm. Under this assumption, different variants of GD are analyzed, including GD with clipping (Clip-GD) (Zhang et al., 2020b;a; Koloskova et al., 2023; Vankov et al., 2024a), $(L_0, L_1)$-GD (Gorbunov et al., 2024; Vankov et al., 2024a), Normalized GD (NGD) (Zhao et al., 2021; Chen et al., 2023; Vankov et al., 2024a), and other variants (Crawshaw et al., 2022; Wang et al., 2022; Faw et al., 2023; Wang et al., 2023; Hübler et al., 2024; Li et al., 2024b). More precisely, in the deterministic convex case, the state-of-the-art results for Clip-GD, $(L_0, L_1)$-GD, and NGD are obtained by Gorbunov et al. (2024); Vankov et al. (2024a) showing the $\mathcal{O}\left( \frac{L_0 R^2}{N} \right)$ rates for function suboptimality when $N = \Omega(L_1^2 R^2)$[1] leaving open questions about the refined methods behavior characterization for $N = \mathcal{O}(L_1^2 R^2)$.

---

[1]After we derived the results of this paper we learned that Vankov et al. (2024b) independently derived $\mathcal{O}\left( \frac{L_0 R^2}{N} + \left( 1 - \frac{1}{L_1 R} \right)^N F_0 \right)$ rate for $(L_0, L_1)$-GD, where $F_0 = f(x^0) - f(x^*)$ and updated their paper on

However, beyond the first-order methods, the algorithms for $(L_0, L_1)$-smooth optimization are weakly studied. In particular, *random coordinate descent* (RCD) Nesterov (2012); Shalev-Shwartz & Tewari (2009); Richtárik & Takáč (2016), which is useful in the situations when the computation of the full gradient is prohibitively expensive, is not analyzed in the context of $(L_0, L_1)$-smooth optimization. Moreover, in some cases, e.g., in the reinforcement learning with human feedback (Tang et al., 2024), even objective values are available, and for given points $x, y \in \mathbb{R}^d$ one can only evaluate $\text{sign}[f(y) - f(x)]$. To the best of our knowledge, there are no theoretical convergence results for such methods under $(L_0, L_1)$-smoothness, and, in particular, the convergence of *random coordinate descent with order oracle* (OrderRCD) (Lobanov et al., 2024) is not studied in this setup.

In this paper, we address this gap in the literature and provide the first analysis of RCD and OrderRCD for convex $(L_0, L_1)$-smooth optimization. Moreover, we improve the existing results for $(L_0, L_1)$-GD, NGD, and Clip-GD: we prove that these methods enjoy *exponential objective decrease without any additional assumptions* for the initial optimization phase when $\left\| \nabla f(x^k) \right\| \geq \frac{L_0}{L_1}$ (see red line in Figure 1). Figure 1 demonstrates the changing regimes (from linear to sublinear) through the theoretical threshold $\left\| \nabla f(x^k) \right\| = \frac{L_0}{L_1}$ using the example of $(L_0, L_1)$-GD convergence. For a more detailed description of Figure 1, see the Numerical Experiments section.

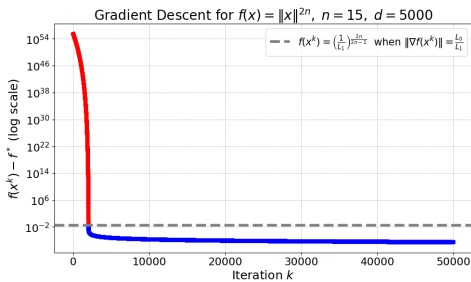

Figure 1: Changing regimes demonstration

Our contributions can be summarized as follows.

- We show under what conditions the gradient descent variants exhibit exponential objective decrease in the convex setup (when $\left\| \nabla f(x^k) \right\| \geq \frac{L_0}{L_1}$, including the case $L_0 = 0$).

- We prove better complexity bounds for $(L_0, L_1)$-GD, NGD, and Clip-GD than previously known ones, assuming convexity and $(L_0, L_1)$-smoothness of the objective function. We show that these algorithms exhibit a fast initial convergence phase (exponential objective decrease), and slow down as they approach the solution, converging sublinearly. We also show that for the phase of convergence of NGD, when the iterates satisfy $\left\| \nabla f(x^k) \right\| \geq c$, the method converges linearly. Table 1 demonstrates the conditions under which Clip-GD exhibits exponential objective decrease. In particular, the case of $\lambda_k = 1$ corresponds to the convergence of GD, and the case of $\lambda_k = \frac{c}{\|\nabla f(x^k)\|}$ corresponds to the convergence of NGD.

- We provide the first convergence results for the RCD and OrderRCD algorithms under the convexity and $(L_0, L_1)$-coordinate smoothness assumptions. We show that fast initial convergence phenomenon of the full-gradient methods exists for both of the mentioned methods.

- We extend our analysis of $(L_0, L_1)$-GD to the case when the function is $\mu$-strongly convex.

### 1.1 NOTATIONS AND MAIN ASSUMPTIONS

Before discussing related work, we introduce the notations and assumptions that are used in this paper.

**Notations.** We use $\langle x, y \rangle := \sum_{i=1}^d x_i y_i$ to denote standard inner product of $x, y \in \mathbb{R}^d$. We denote Euclidean norm in $\mathbb{R}^d$ as $\|x\| := \sqrt{\sum_{i=1}^d x_i^2} = \sqrt{\langle x, y \rangle}$. We use $\mathbf{e}_i \in \mathbb{R}^d$ to denote the $i$-th unit vector. For $\boldsymbol{L} = (L^{(1)}, \ldots, L^{(d)})^\top \in \mathbb{R}^d$ and $\alpha \in \mathbb{R}$, we define the norms $\|x\|_{[\boldsymbol{L}, \alpha]} := \sqrt{\sum_{i=1}^d (L^{(i)})^\alpha x_i^2}$ and $\|x\|_{[L_p, \alpha]}^* := \sqrt{\sum_{i=1}^d \frac{1}{(L_p^{(i)})^\alpha} x_i^2}$. We denote by $\nabla f(x)$ the full gradient of function $f$ at point $x \in \mathbb{R}^d$, and by $\nabla_i f(x)$ the $i$-th coordinate gradient of function $f$ at point $x \in \mathbb{R}^d$. We also introduce $S_\alpha^{\boldsymbol{L}} := \sum_i^d (L^{(i)})^\alpha$. We use $\tilde{O}(\cdot)$ to hide the logarithmic coefficients. We denote $f^* := f(x^*)$ and $x^* \in X^* := \arg\min_{x \in \mathbb{R}^d} f(x)$ to be any solution of equation 1. We also use $R := \|x^0 - x^*\|$ and $F_0 := f(x^0) - f^*$.

The standard smoothness assumption (see, e.g., Nesterov, 2013) is $L$-smoothness.

arXiv (Vankov et al., 2024b). At the moment of writing our paper, we were unaware of the updated version of (Vankov et al., 2024b).

Table 1: Comparison of the convergence rates for Clip-GD in the convex case. Clip-GD update scheme: $x^{k+1} = x^k - \eta_k \cdot \mathrm{clip}_c(\nabla f(x^k))$. Notation: $\mathrm{clip}_c(\nabla f(x^k)) = \lambda_k \cdot \nabla f(x^k)$; $\lambda_k = \min\{1, {}^c\!/\|\nabla f(x^k)\|\}$; $c > 0$ – clipping radius; $\eta_k > 0$ – step size; $N$ = number of iterations; $F_0 = f(x^0) - f^*$; $R = \|x^0 - x^*\|$; $T = \min\{k \in \{0, 1, ..., N-1\} \mid \|\nabla f(x^k)\| < L_0/L_1\}$; SEOD = summand with exponential objective decrease.

| Reference | Clipping threshold | $\lambda_k$ | Smoothness case: $L_0$ (?) $cL_1$ | Convergence rate $f(x^N) - f^* \lesssim$ | SEOD? |
|---|---|---|---|---|---|
| Koloskova et al. (2023) | arbitrary | 1 | larger | $\mathcal{O}\left(\frac{L_0 R^2}{N}\right)$ | ✗ |
| | | | less or equal | $\mathcal{O}\left(\frac{cL_1 R^2}{N}\right)$ | ✗ |
| | | $\frac{c}{\|\nabla f(x^k)\|}$ | larger | $\mathcal{O}\left(\frac{L_0^2 L R^4}{c^2 N^2}\right)$ | ✗ |
| | | | less or equal | $\mathcal{O}\left(\frac{L_1^2 L R^4}{N^2}\right)$ | ✗ |
| Gorbunov et al. (2024) & Vankov et al. (2024a) | $c = \frac{L_0}{L_1}$ | 1 | equal | $\mathcal{O}\left(\frac{L_0 R^2}{N}\right)$ | ✗ |
| | | $\frac{c}{\|\nabla f(x^k)\|}$ | equal | ✗ | ✗ |
| **Theorem 3.5 (Our work)** | arbitrary | 1 | larger | $\mathcal{O}\left(\frac{L_0 R^2}{N}\right)$ | ✗ |
| | | | less or equal | $\mathcal{O}\left(\min\left\{\frac{L_0 R^2}{N-T}, \left(1 - \frac{1}{L_1 R}\right)^T F_0\right\}\right)$ | ✓ |
| | | $\frac{c}{\|\nabla f(x^k)\|}$ | larger | $\mathcal{O}\left(\left(1 - \frac{c}{L_0 R}\right)^N F_0\right)$ | ✓ |
| | | | less or equal | $\mathcal{O}\left(\left(1 - \frac{1}{L_1 R}\right)^N F_0\right)$ | ✓ |

**Assumption 1.1** ($L$-smoothness). Function $f$ is $L$-smooth if the following inequality is satisfied for any $x, y \in \mathbb{R}^d$:

$$\|\nabla f(y) - \nabla f(x)\| \leq L \|y - x\|.$$

However, instead of standard $L$-smoothness, we focus on the so-called $(L_0, L_1)$-smoothness (Zhang et al., 2020b;a).

**Assumption 1.2** ($(L_0, L_1)$-smoothness). Function $f : \mathbb{R}^d \to \mathbb{R}$ is $(L_0, L_1)$-smooth if the following inequality is satisfied for any $x, y \in \mathbb{R}^d$ with $\|y - x\| \leq \frac{1}{L_1}$:

$$\|\nabla f(y) - \nabla f(x)\| \leq (L_0 + L_1 \|\nabla f(x)\|) \|y - x\|. \tag{2}$$

If $L_1 = 0$, the above assumption recovers Assumption 1.1 with $L = L_0$. Moreover, $(L_0, L_1)$-smoothness is strictly more general than $L$-smoothness, see the examples in Zhang et al. (2020b); Chen et al. (2023); Koloskova et al. (2023); Gorbunov et al. (2024).

**Remark 1.3.** *In this paper we often emphasize the case $L_0 = 0$ in Assumption 1.2. It is worth noting that the class of functions that do not reach their infimum $x^*$ (converge to an asymptote) satisfies this case. Explicit examples of functions with $L_0 = 0$ are the exponent of the inner product and the logistic function (see Gorbunov et al. (2024) for details).*

Next, we also use a coordinate-wise version of Assumption 1.2 introduced by Crawshaw et al. (2022).

**Assumption 1.4** ($(L_0, L_1)$-coordinate-smoothness). A function $f : \mathbb{R}^d \to \mathbb{R}$ is $(L_0, L_1)$-coordinate-smooth for $L_0^{(1)}, L_0^{(2)}, ..., L_0^{(d)}, L_1^{(1)}, L_1^{(2)}, ..., L_1^{(d)} \geq 0$) if for any $i \in [d]$, $x \in \mathbb{R}^d$ and $h \in \mathbb{R}$, $|h| \leq \frac{1}{\max_{i \in [d]} L_1^{(i)}}$ the following inequality holds:

$$|\nabla_i f(x + h\mathbf{e}_i) - \nabla_i f(x)| \leq \left(L_0^{(i)} + L_1^{(i)} |\nabla_i f(x)|\right) |h|.$$

The above assumption generalizes the standard coordinate $L$-smoothness (Lin et al., 2014; Allen-Zhu et al., 2016; Zhang & Xiao, 2017) similarly to how $(L_0, L_1)$-smoothness generalizes $L$-smoothness.

We also assume that the function $f$ is ($\mu$-strongly) convex.

**Assumption 1.5.** Function $f : \mathbb{R}^d \to \mathbb{R}$ is $\mu \geq 0$ strongly convex if for any $x, y \in \mathbb{R}^d$ the following inequality holds:

$$f(y) \geq f(x) + \langle \nabla f(x), y - x \rangle + \frac{\mu}{2} \|y - x\|^2. \tag{3}$$

Assumption 1.5 is classical and widely used in the literature (see, e.g., Boyd & Vandenberghe, 2004; Nesterov, 2018).

## 1.2 PAPER STRUCTURE

Further, our paper has the following structure. In Section 2, we discuss the related work. In Section 3, we provide the results for full-gradient methods. The case where the oracle only has access to the gradient coordinate or the comparison of the values of two functions is considered in Section 4. In Section 5, we generalize our results for $(L_0, L_1)$-GD to the strongly convex case. Discussions of this work and future work plans are given in Section 6. In Section 7, we confirm our theoretical results via numerical experiments. Section 8 concludes the paper. All proofs are provided in the Appendix.

## 2 RELATED WORKS

The literature on the analysis of GD-type methods is very rich. We discuss only closely related works.

**Full-gradient methods for the $(L_0, L_1)$-smooth convex optimization.** Although most of the existing works on $(L_0, L_1)$-smoothness focus on the non-convex case, there are several papers considering the (strongly) convex problems as well. Koloskova et al. (2023) gives the first analysis Clip-GD (Pascanu et al., 2013) under $(L_0, L_1)$-smoothness and $L$-smoothness and proves $\mathcal{O}\left(\max\left\{\frac{(L_0+cL_1)R^2}{N}, \frac{R^4 L(L_0+cL_1)^2}{c^2 N^2}\right\}\right)$ rate (see Table 1 for the details). This bound is derived under the additional $L$-smoothness assumption, which is not always satisfied for $(L_0, L_1)$-smooth problems. Moreover, when $\lambda_k = 1$ and $L_0 \leq cL_1$, the derived rate is proportional to $c$. In addition, the analysis from (Koloskova et al., 2023) implies the sublinear convergence rate for NGD (see the case $\lambda_k = \frac{c}{\|\nabla f(x^k)\|}$ in Table 1). Takezawa et al. (2024) derive similar results for GD with Polyak Stepsizes (GD-PS), i.e., they show $\mathcal{O}\left(\max\left\{\frac{L_0 R^2}{N}, \frac{R^4 L L_1^2}{c^2 N^2}\right\}\right)$ convergence rate. Next, Li et al. (2024a) derive convergence rates for GD and its accelerated version under $(r, \ell)$-smoothness assumption, which generalizes $(L_0, L_1)$-smoothness. In particular, for GD Li et al. (2024a) prove $\mathcal{O}(\frac{\ell R^2}{N})$ convergence rate, where $\ell = \mathcal{O}(L_0 + L_1 G)$ and constant $G$ depends in $L_0, L_1, R, \|\nabla f(x^0)\|$, and $f(x^0) - f^*$, meaning that it can be exponentially large in terms of $L_1$ and $R$. Finally, Gorbunov et al. (2024); Vankov et al. (2024a) independently improve the convergence rates of $(L_0, L_1)$-GD/Clip-GD by considering the special case of clipping radius $c = \frac{L_0}{L_1}$. More precisely, they prove $\mathcal{O}\left(\frac{L_0 R^2}{N}\right)$ convergence rate if $N \geq L_1^2 R^2$ and extend this result to GD-PS (Vankov et al. (2024a) also show a similar result for NGD). However, the results from Gorbunov et al. (2024); Vankov et al. (2024a) do not provide convergence rates in terms of $f(x^N) - f^*$ for the stage when $\|\nabla f(x^k)\| > L_0/L_1$, which can be noticeable when $L_0$ is small and $L_1$ is large. *In our work, we propose the analysis that addresses this limitation (see Table 1).*

**Coordinate descent (CD) type methods.** Convergence of coordinate methods is also relatively well-studied. For example, under the standard $L$-smoothness assumption $((\nabla^2 f(x))_{i,i} \leq L)$, the CD method has the following convergence rate $\mathcal{O}\left(\frac{dL R^2}{N}\right)$ (see, e.g., Bubeck et al., 2015). Using the fact that $\frac{1}{d}\sum_{i=1}^d L^{(i)} \leq L$ and assuming $L$-coordinate-smoothness $((\nabla^2 f(x))_{i,i} \leq L^{(i)})$, the previous result can be improved to $\mathcal{O}\left(\frac{\sum_{i=1}^d L^{(i)} R^2}{N}\right)$ rate. Next, assuming that the active coordinate $i_k$ can be obtained (independently) from the distribution $p_\alpha(i) = (L^{(i)})^\alpha/S_\alpha$, then RCD converges at $\mathcal{O}\left(\frac{S_\alpha R_{[L,1-\alpha]}^2}{N}\right)$ rate Nesterov (2012), where $R_{[L,1-\alpha]} := \max_{x \in \mathbb{R}^d}\left\{\max_{x^* \in X^*} \|x - x^*\|_{[L,1-\alpha]} : f(x) \leq f(x^0)\right\}$. Moreover, Lobanov et al. (2024) show that it is possible to create an OrderRCD algorithm based on RCD, whose oracle has access only to function comparisons (this oracle can be motivated by, e.g., RLHF (Ouyang et al., 2022; Bai et al., 2022)). More precisely, Lobanov et al. (2024) prove that the iteration complexity of OrderRCD is the same as for RCD, and the oracle complexity is inferior only in $\log(1/\epsilon)$ factor, where $\epsilon$ is the accuracy of the solution of linear search problem. *In our paper, we extend these results to the more general case.*

## 3 FULL-GRADIENT METHODS

In this section, we present our result for full-gradient algorithms (see GD in Subsection 3.1, NGD in Subsection 3.2, and Clip-GD in Subsection 3.3).

## 3.1 GRADIENT DESCENT METHOD

The first algorithm we consider is the gradient descent method (see Algorithm 1). We prove the following result for Algorithm 1 with stepsize $\eta_k = (L_0 + L_1 \|\nabla f(x^k)\|)^{-1}$ (to emphasize the specificity of the step size we call it $(L_0, L_1)$-GD for brevity). In Theorem 3.1, we consider in detail two cases of convergence depending on $\|\nabla f(x^k)\|$, and provide the general rate

---
**Algorithm 1** Gradient Descent Method (GD)

---
**Input:** $x_0 \in \mathbb{R}^d$, iterations number $N$, step size $\eta_k > 0$
**for** $k = 0$ to $N - 1$ **do**
    $x^{k+1} \leftarrow x^k - \eta_k \nabla f(x^k)$
**end for**
**Return:** $x^N$

---

of convergence of $(L_0, L_1)$-gradient descent method (see Algorithm 1) in the convex setup.

**Theorem 3.1.** *Let function $f$ satisfy Assumption 1.2 ($(L_0, L_1)$-smoothness) and Assumption 1.5 (convexity, $\mu = 0$), then GD (Algorithm 1) with step size $\eta_k = (L_0 + L_1 \|\nabla f(x^k)\|)^{-1}$ guarantees*

- *exponential objective decrease, if $\|\nabla f(x^k)\| \geq \frac{L_0}{L_1}$ for $k \in [N-1]$*

$$f(x^N) - f^* \leq \left(1 - \frac{1}{4L_1 R}\right)^N (f(x^0) - f^*);$$

- *sublinear convergence, if $\|\nabla f(x^{N-1})\| < \frac{L_0}{L_1}$:*

$$f(x^N) - f^* < \frac{4L_0 R^2}{N}.$$

*In the general case, the convergence rate is*

$$f(x^N) - f^* \leq \min\left\{\frac{4L_0 R^2}{N - T}, \left(1 - \frac{1}{4L_1 R}\right)^T F_0\right\},$$

*where $T \geq 0$ is the smallest index such as $\|\nabla f(x^T)\| < \frac{L_0}{L_1}$.*

Given the monotonicity of the gradient norm (see Appendix B), Theorem 3.1 characterizes in details the convergence behavior of GD for convex $(L_0, L_1)$-smooth problems. More precisely, as long as the gradient norm is larger than $L_0/L_1$, GD shows exponential objective decrease, but when the method approaches the solution ($\|\nabla f(x^k)\| < L_0/L_1$) the convergence slows down to the standard sublinear rate. That is, $\mathcal{O}(\frac{L_0 R^2}{N})$ rate is common to the previous works (Gorbunov et al., 2024; Vankov et al., 2024a) (see Table 1). However, in contrast to (Gorbunov et al., 2024; Vankov et al., 2024a), our analysis shows $\mathcal{O}(1 - 1/L_1 R)^N F_0$ rate when $\|\nabla f(x^k)\| \geq L_0/L_1$ (see the case of $\lambda_k = c/\|\nabla f(x^k)\|$ in Table 1). Moreover, our result improves the one from (Koloskova et al., 2023) (see smoothness case "less or equal" with $\lambda_k = 1$ in Table 1) from sublinear summand to SEOD and gets rid of potentially large parameter $c \geq L_0/L_1$. The proof of the Theorem 3.1 is provided in Appendix C.1.

The significance of the improved estimate can be observed when $L_0 = 0$ (see Remark 1.3).

**Remark 3.2.** *Theorem 3.1 implies that under Assumption 1.2 with $L_0 = 0$, Algorithm 1 converges to the desired accuracy $\varepsilon$ ($f(x^N) - f^* \leq \varepsilon$) after $N = \mathcal{O}\left(L_1 R \log \frac{F_0}{\varepsilon}\right)$ iterations.*

The result of Remark 3.2 significantly outperforms all known results in this regime. In particular, Koloskova et al. (2023) show $\mathcal{O}(L_1 c R^2/\varepsilon)$ complexity bound for Clip-GD, and Gorbunov et al. (2024); Vankov et al. (2024a) do not provide explicit rates in this case.

## 3.2 NORMALIZED GD METHOD

From the previous section, we see that GD with step size with step size $\eta_k = (L_0 + L_1 \|\nabla f(x^k)\|)^{-1}$ enjoys exponential objective decrease in convex setup, when $\|\nabla f(x^k)\| \geq L_0/L_1$. However, in this regime, we have $L_1 \|\nabla f(x^k)\| \geq L_0$, meaning that $(2L_1 \|\nabla f(x^k)\|)^{-1} \leq \eta_k \leq (L_1 \|\nabla f(x^k)\|)^{-1}$, i.e., the method is very close to NGD (Algorithm 2). Therefore, it is natural to expect similar behavior from NGD as for $(L_0, L_1)$-GD. The following result formalizes this observation (see details Theorem 3.3).

---

**Algorithm 2** Normalized Gradient Descent Method (NGD)

**Input:** $x_0 \in \mathbb{R}^d$, iterations number $N$, step size $\eta_k > 0$
**for** $k = 0$ to $N - 1$ **do**
  **if** $\|\nabla f(x^k)\| = 0$ **then**
    **Return:** $x^k$
  **end if**
  $x^{k+1} \leftarrow x^k - \eta_k \frac{\nabla f(x^k)}{\|\nabla f(x^k)\|}$
**end for**
**Return:** $x^N$

---

**Theorem 3.3.** *Let function $f$ satisfy Assumption 1.2 ($(L_0, L_1)$-smoothness) and Assumption 1.5 (convexity, $\mu = 0$), then Algorithm 2 with step size $\eta_k = \eta \leq c/(L_0 + L_1 c)$, where constant $c > 0$ is such that $\|\nabla f(x^k)\| \geq c$ for all $k = 0, 1, \ldots, N - 1$, has linear convergence:*

$$ f(x^N) - f^* \leq \left(1 - \frac{\eta}{2R}\right)^N (f(x^0) - f^*). $$

Theorem 3.3 shows that in the case of $c \geq L_0/L_1$, NGD has $\mathcal{O}\left((1 - \frac{1}{L_1 R})^N F_0\right)$ convergence rate similarly to GD, which is natural to expect due to $\|\nabla f(x^k)\| \geq c$ and the discussion given in the beginning of this subsection. However, if we select large enough $N$, one has to select $c$ small enough such that $\|\nabla f(x^k)\| \geq c$ holds for all $k = 0, 1, \ldots, N - 1$. If $c < L_0/L_1$, then the rate reduces to $\mathcal{O}\left((1 - c/(L_0 R))^N F_0\right)$ and the method is guaranteed to converge only to the error $\varepsilon \sim cR$. Therefore, to guarantee the convergence to $\varepsilon$-accuracy, one has to take $c \sim \varepsilon/R$ in the worst case. In this case, our result implies $\mathcal{O}(L_0 R^2 \log(F_0/\varepsilon)/\varepsilon)$ complexity for NGD. However, hyperparameter $c$ depends only on the gradient norm, so in problems where the high accuracy on gradient norm is not required, Algorithm 2 is efficient and shows linear convergence. The proof of Theorem 3.3 see Appendix C.2.

**Remark 3.4.** *Theorem 3.3 implies that under Assumption 1.2 with $L_0 = 0$, Algorithm 2 converges to the desired accuracy $\varepsilon$ ($f(x^N) - f^* \leq \varepsilon$) after $N = \mathcal{O}\left(L_1 R \log \frac{F_0}{\varepsilon}\right)$ iterations.*

As previously noted, when $\|\nabla f(x^k)\| \geq \frac{L_0}{L_1}$ $(L_0, L_1)$-GD and NGD with $c \geq \frac{L_0}{L_1}$ are almost the same. Therefore, the result of Remark 3.4 is expected given Remark 3.2. Moreover, our results imply that NGD has $\mathcal{O}(\max\{L_0 R^2 \log(F_0/\varepsilon)/\varepsilon, L_1 R \log(F_0/\varepsilon)\})$ complexity. Compared to $\mathcal{O}(\max\{L_0 \overline{R}^2/\varepsilon, L_1^2 \overline{R}^2\})$ complexity bound derived for NGD with $\eta_k = \hat{R}/\sqrt{N+1}$, $\overline{R} := \hat{R} + R^2/\hat{R}$ by Vankov et al. (2024a), our bound has an additional logarithmic factor in the first term but has much better second term when $L_1 R$ is large and $\log(F_0/\varepsilon)$ is much smaller than $L_1 R$.

## 3.3 CLIPPED GD METHOD

In this section, we consider Clip-GD (Algorithm 3), which applies the clipping operator to the gradient:

$$ \text{clip}_c(\nabla f(x)) = \lambda_k \cdot \nabla f(x), \quad (4) $$

where $\lambda_k = \min\{1, c/\|\nabla f(x^k)\|\}$, $c > 0$ is the clipping radius. Clip-GD can also be seen as a combination of GD (when $\|\nabla f(x^k)\| \leq c$) and NGD (when $\|\nabla f(x^k)\| > c$). Then, following similar reasoning as in the previous sections,

---

**Algorithm 3** Clipped Gradient Descent Method (Clip-GD)

**Input:** initial point $x_0 \in \mathbb{R}^d$, iterations number $N$, step size $\eta_k > 0$ and clipping radius $c > 0$
**for** $k = 0$ to $N - 1$ **do**
  $x^{k+1} \leftarrow x^k - \eta_k \cdot \text{clip}_c(\nabla f(x^k))$ according to equation 4
**end for**
**Return:** $x^N$

---

we obtain the next convergence result for Clip-GD method (see more details Theorem 3.5).

**Theorem 3.5.** *Let function $f$ satisfy Assumption 1.2 ($(L_0, L_1)$-smoothness) and Assumption 1.5 (convexity, $\mu = 0$), then Algorithm 3 with step size $\eta_k = (L_0 + L_1 \min\{\|\nabla f(x^k)\|, c\})^{-1}$ guarantees:*

$$f(x^N) - f^* = \mathcal{O}\left(\min\left\{\frac{L_0 R^2}{N - T}, \left(1 - \frac{\rho}{R}\right)^T F_0\right\}\right),$$

*where $\rho := {c}/{\max\{L_0, L_1 c\}}$ and $T \geq 0$ is the smallest index such as $\|\nabla f(x^T)\| < \min\{c, {L_0}/{L_1}\}$*

Since NGD and GD are monotonically decreasing in terms of the gradient norm, it follows that Algorithm 3 is also monotonically decreasing in terms of the gradient norm (see Appendix B for details). Given this fact, Theorem 3.5 shows that Algorithm 3 has two convergence regimes depending on the ratio of $c$ and ${L_0}/{L_1}$. If $c \geq {L_0}/{L_1}$, then Clip-GD starts its convergence with exponential objective decrease $\mathcal{O}\left((1 - ({1}/{L_1 R}))^N F_0\right)$, and as soon as it approaches the solution, i.e., when $\|\nabla f(x^k)\| < {L_0}/{L_1}$, it slows down to a sublinear $\mathcal{O}\left({L_0 R^2}/{N}\right)$ rate. If $c < {L_0}/{L_1}$, then Clip-GD has inferior exponential objective decrease $\mathcal{O}\left((1 - ({c}/{L_0 R}))^N F_0\right)$ at the beginning, and approaching the solution, i.e., when $\|\nabla f(x^k)\| < c$, it slows down to the same sublinear rate. The cases in Appendix C.3 are discussed in more detail. Table 1 summarizes the derived results and compares them with the closely related works analyzing Clip-GD. It is worth noting that Theorem 3.5 shows when Algorithm 3 has exponential objective decrease and gets rid of standard smoothness constant $L$ (in contrast to (Koloskova et al., 2023)). Moreover, Theorem 3.5 is valid for an arbitrary clipping threshold $c$ (in contrast to (Gorbunov et al., 2024; Vankov et al., 2024a)).

**Remark 3.6.** *Theorem 3.5 implies that under Assumption 1.2 with $L_0 = 0$, Algorithm 3 converges to the desired accuracy $\varepsilon$ ($f(x^N) - f^* \leq \varepsilon$) after $N = \mathcal{O}\left(L_1 R \log \frac{F_0}{\varepsilon}\right)$.*

## 4 COORDINATE DESCENT TYPE METHODS

In this section, we present our main results for the algorithms that does not use access to the full gradient (see RCD in Subsection 4.1, and OrderRCD see Subsection 4.2).

### 4.1 RANDOM COORDINATE DESCENT

Random coordinate descent is formalized as Algorithm 4. At each iteration, the method computes the gradient coordinate $\nabla_{i_k} f(x^k)$, where active coordinate $i_k$ is selected uniformly at random from $[d]$ independently from previous steps. The main difference from previous results is that due to the randomness of the choice of the active coordinate, RCD is not monotonic in terms of the gradient norm. This challenge has been addressed in our main results for RCD are given below.

---

**Algorithm 4** Random Coordinate Descent Method (RCD)

**Input:** initial point $x_0 \in \mathbb{R}^d$, iterations number $N$, step size $\eta_k > 0$
**for** $k = 0$ **to** $N - 1$ **do**
   1. sample $i_k$ uniformly at random from $[d]$
   2. $x^{k+1} \leftarrow x^k - \eta_k \nabla_{i_k} f(x^k) \mathbf{e}_{i_k}$
**end for**
**Return:** $x^N$

---

**Theorem 4.1.** *Let function $f$ satisfy Assumption 1.4 ($(L_0, L_1)$-coordinate-smoothness) and Assumption 1.5 (convexity, $\mu = 0$), then RCD (Algorithm 4) with step size $\eta_k \leq (L_0 + L_1 |\nabla_{i_k} f(x^k)|)^{-1}$, where $L_0 = \max_{i \in [d]} L_0^{(i)}$ and $L_1 = \max_{i \in [d]} L_1^{(i)}$, guarantees the following error:*

$$\mathbb{E}\left[f(x^N)\right] - f^* = \mathcal{O}\left(\max\left\{\left(1 - \frac{1}{4\sqrt{2} d R L_1}\right)^N F_0, \frac{d L_0 R^2}{N}\right\}\right).$$

Theorem 4.1 provides a generalization of the results of Nesterov (2012) to the case of $(L_0, L_1)$-coordinate-smoothness. In particular, following Section 3, we separated $L_0$ and $L_1$ in the convergence

results and also show that there is no need to assume standard $L$-smoothness since the case $L_1 = 0$ covers it. Moreover, in the case of $L_0$ being much smaller than $L_1$, the results of Theorem 4.1 are strictly better than previously known ones. In the case of $L_0 = 0$, RCD demonstrates exponential objective decrease to any accuracy $\varepsilon$. For a detailed proof of Theorem 4.1, see Appendix D.1.

**Remark 4.2.** *Theorem 4.1 implies that under Assumption 1.4 with $L_0 = 0$, RCD converges to the desired accuracy $\varepsilon$ ($\mathbb{E}[f(x^N)] - f^* \leq \varepsilon$) after $N = \mathcal{O}\left(dL_1 R \log \frac{F_0}{\varepsilon}\right)$ iterations.*

## 4.2 RANDOM COORDINATE DESCENT WITH ORDER ORACLE

In this section, we consider the OrderRCD (Algorithm 5). In contrast to all previously considered methods in this paper, OrderRCD does not have access to a first-order oracle. Instead, the algorithm uses so-called Order Oracle: for any $x, y \in \mathbb{R}^d$, one can compute

$$\psi(x, y) = \text{sign}\left[f(y) - f(x)\right]. \quad (5)$$

Algorithm 5 is similar to Algorithm 4, but it does not have access to the gradient coordinate $\nabla_{i_k} f(x^k)$. Following Lobanov et al. (2024), we address this challenge using the standard steepest descent trick, namely, we solve at each iteration the auxiliary linear search problem

---

**Algorithm 5** RCD with Order Oracle (OrderRCD)

---

**Input:** initial point $x_0 \in \mathbb{R}^d$, iterations number $N$, random generator $\mathcal{R}_\alpha(L_0, L_1)$
**for** $k = 0$ **to** $N - 1$ **do**
   1. sample $i_k$ uniformly at random from $[d]$
   2. compute $\zeta_k = \text{argmin}_\zeta\{f(x^k + \zeta\mathbf{e}_{i_k})\}$ via (GRM)

   3. $x^{k+1} \leftarrow x^k + \zeta_k\mathbf{e}_{i_k}$
**end for**
**Return:** $x^N$

---

using the golden ratio method (see Algorithm 6 in Appendix D.2, GRM) with $\epsilon$ accuracy allowing to match Random coordinate descent with step size $\eta_k$.

Below, we present the convergence result for Algorithm 5.

**Theorem 4.3.** *Let function $f$ satisfy Assumption 1.4 (($L_0, L_1$)-coordinate-smoothness) and Assumption 1.5 (convexity, $\mu = 0$), then Algorithm 5 (OrderRCD) with oracle equation 5 guarantees:*

$$\mathbb{E}\left[f(x^N)\right] - f^* = \mathcal{O}\left(\max\left\{\left(1 - \frac{\rho}{dR}\right)^N F_0, \frac{dL_0 R^2}{N}\right\}\right),$$

*where $\rho := 1/(4\sqrt{2}L_1)$, $L_0 = \max_{i \in [d]} L_0^{(i)}$, and $L_1 = \max_{i \in [d]} L_1^{(i)}$.*

That is, Theorem 4.3 gives exactly the same rate as Theorem 4.1 with one exception. It is important to note that Algorithm 5 requires $\log(1/\epsilon)$ oracle calls per iteration to solve the linear search problem at each iteration using GRM, where Order Oracle equation 5 is directly used. In the special case of $L_1 = 0$, Theorem 4.3 recovers known results, e.g., Gorbunov et al. (2019); Saha et al. (2021). However, when $L_0$ is much smaller than $L_1$, Theorem 4.3 shows better results, i.e., exponential objective decrease (see Remark 4.4). For a detailed proof, see Appendix D.2.

**Remark 4.4.** *Theorem 4.3 implies that under Assumption 1.4 with $L_0 = 0$, Algorithm 5 converges to the desired accuracy $\varepsilon$ ($\mathbb{E}[f(x^N)] - f^* \leq \varepsilon$) after $N = \mathcal{O}\left(dL_1 R \log \frac{F_0}{\varepsilon}\right)$ iterations and $T = \mathcal{O}\left(N \log \frac{1}{\epsilon}\right)$ oracle calls.*

## 5 EXTENSION TO STRONGLY CONVEX SETUP

In this section, we answer the question:

*"Are there convergence improvements of algorithms under the $(L_0, L_1)$-smoothness assumption compared to standard smoothness in a strongly convex setup?"*

In particular, we consider $(L_0, L_1)$-GD (Algorithm 1) and derive the following convergence result.

**Theorem 5.1.** *Let function $f$ satisfy Assumption 1.2 ($(L_0, L_1)$-smoothness) and Assumption 1.5 (strongly convexity, $\mu > 0$), then $(L_0, L_1)$-GD with $\eta_k = (L_0 + L_1 \|\nabla f(x^k)\|)^{-1}$ guarantees:*

$$F_N \leq (1 - \rho_3)^{N - \mathcal{T}_2} (1 - \rho_2)^{\mathcal{T}_2 - \mathcal{T}_1} (1 - \rho_1)^{\mathcal{T}_1 + 1} F_0.$$

*where $F_k = f(x^k) - f^*$ for $k \in [N]$, $\rho_3 = \frac{\mu}{2L_0}$, $\rho_2 = \max\left\{\frac{\sqrt{\mu}}{2\sqrt{2}L_1}, \frac{1}{4L_1 R}\right\}$, $\rho_1 = \frac{1}{4L_1 R}$, $N_3 = N - \mathcal{T}_2$, $\mathcal{T}_2 = \max\{k \in [N-1] \mid \|\nabla f(x^k)\| \geq \frac{L_0}{L_1}\}$ (if there are no such $k$, we let $\mathcal{T}_2 = -1$), and $\mathcal{T}_1 = \max\{k \in [N-1] \mid \|\nabla f(x^k)\| \geq \frac{L_0}{L_1}$ and $F_k > 1\}$ (if there are no such $k$, we let $\mathcal{T}_1 = -1$).*
The above theorem improves the result from (Gorbunov et al., 2024) that show $\|x^N - x^*\|^2 = \mathcal{O}((1 - \rho_3)^{N - \mathcal{T}_2} R)$ rate for GD, and, in contrast to the result from (Koloskova et al., 2023), Theorem 5.1 does not require $L$-smoothness. Moreover, the derived bound contains factor $(1 - \rho_2)^{\mathcal{T}_2 - \mathcal{T}_1}(1 - \rho_1)^{\mathcal{T}_1 + 1}$, which might be better than $(1 - \rho_1)^{\mathcal{T}_2}$ when $R > \sqrt{2/\mu}$. Moreover, if $\mu/2L_0 < \sqrt{\mu}/(2\sqrt{2}L_1)$, then the derived result is strictly better than the known ones for GD under the standard smoothness.

## 6 DISCUSSION AND FUTURE WORK

In the Sections 3 and 4 we have shown that exponential objective decrease in a convex setup is possible in the case of $(L_0, L_1)$-smooth problems with small enough $L_0$. However, looking at the convergence of Algorithms 1-5, in particular Theorem 3.1-4.3, we see that the dominant part is sublinear $\mathcal{O}(1/N)$. Nevertheless, as Remarks 3.2-4.4 demonstrate, in the case $L_0 = 0$, we can observe significant improvements compared to previous works (see Table 1). It is worth noting that functions satisfying the condition $L_0 = 0$ in Assumption 1.2 exist (see Remark 1.3).

As future work, we see the following directions: generalizing Algorithms 2-5 to the strongly convex case; investigating whether the proposed technique can be used in the analysis of stochastic methods. We believe this work opens up a number of research directions, including answering the question of whether it is possible to create adaptive methods, as well as methods in different settings (such as federated learning, overparameterization, etc.) that will exhibit similar advantages.

## 7 NUMERICAL EXPERIMENTS

In this section, we will confirm our theoretical results on the example of functions for which constants $L_0$ and $L_1$ are known. Figure 1 demonstrates the convergence of the $(L_0, L_1)$-GD algorithm on the power of norm problem $f(x) = \|x\|^{2n}$. For this problem, it is known that $L_0 = 2n$ and $L_1 = 2n - 1$. During the tests, the following parameters were taken: $d = 5 \cdot 10^3$; $n = 15$. As we can see in Figure 1, $(L_0, L_1)$-GD has two convergence regimes: exponential objective decrease (see red line), sublinear rate (see blue line). Moreover, the switch between the regimes is carried out exactly along the theoretical

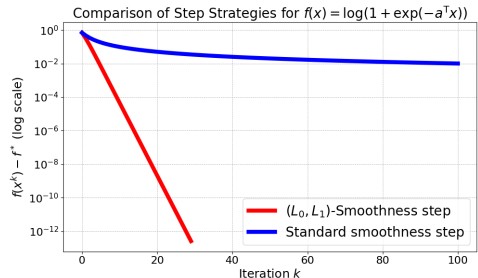

Figure 2: Linear convergence demonstratio

threshold, thus confirming the results presented in the paper above. Figure 2 demonstrates the convergence of GD on the logistic function. We chose this problem because both $L_0 = 0$ and $L_1 = \|a\|$ as well as $L = \|a\|^2$ are known. As we can see in Figure 2, GD under standard smoothness is expected to have a sublinear rate, which is consistent with the lower bounds Nesterov (2018), while $(L_0, L_1)$-GD surprisingly exhibits linear rate, confirming the theoretical results presented in Remarks 3.2-4.4.

## 8 CONCLUSION

This paper demonstrates that generalized smoothness allows us to achieve exponential objective decrease in convex setups. We explained the convergence behavior of gradient descent theoretically and showed that the advantages of generalized smoothness extend to gradient descent method variants, in particular, we significantly improved convergence estimates for GD, NGD, Clip-GD, and demonstrated novel convergence results for algorithms that do not have access to the full gradient as well as to the function values themselves (RCD and OrderRCD). We have demonstrated that this work opens up a number of directions for future research. In Section 7, we confirmed our theoretical results.

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
