# APPENDIX

# Exponential Objective Decrease in Convex Setup is Possible! Gradient Descent Method Variants under $(L_0, L_1)$-Smoothness

## A  AUXILIARY RESULTS

In this section, we provide auxiliary technical results that are used in our analysis.

**Basic inequalities.**  For all $a, b \in \mathbb{R}^d$ ($d \geq 1$), the following inequalities hold:

$$2 \langle a, b \rangle - \|b\|^2 = \|a\|^2 - \|a - b\|^2, \tag{6}$$

$$\langle a, b \rangle \leq \|a\| \cdot \|b\|. \tag{7}$$

**Generalized-Lipschitz-smoothness.**  In the analysis of full-gradient methods, we assume that the $(L_0, L_1)$-smoothness condition (Assumption 1.2) is satisfied. This inequality can be represented in the equivalent form for any $x, y \in \mathbb{R}^d$ (Zhang et al., 2020a):

$$f(y) - f(x) \leq \langle \nabla f(x), y - x \rangle + \frac{L_0 + L_1 \|\nabla f(x)\|}{2} \|y - x\|^2, \tag{8}$$

where $L_0, L_1 \geq 0$ for any $x \in \mathbb{R}^d$ and $\|y - x\| \leq \frac{1}{L_1}$.

**Generalized-coordinate-Lipschitz-smoothness.**  In the analysis of coordinate-wise methods, we assume that the smoothness condition (Assumption 1.4) is satisfied. This inequality can be represented in the equivalent form (Crawshaw et al., 2022, Lemma 1):

$$f(x + h\mathbf{e}_i) \leq f(x) + h\nabla_i f(x) + \frac{\left(L_0^{(i)} + L_1^{(i)}|\nabla_i f(x)|\right) h^2}{2}, \tag{9}$$

where $L_0^{(1)}, L_0^{(2)}, ..., L_0^{(d)}, L_1^{(1)}, L_1^{(2)}, ..., L_1^{(d)} \geq 0$ for any $i \in [d], x \in \mathbb{R}^d$ and $|h| \leq \frac{1}{L_1^{(i)}}$.

## B  MONOTONICITY OF GRADIENT NORMS

In this section, we give a proof of monotonicity of convergence of algorithms by gradient norm. In particular, see Lemma B.2 for the proof for Algorithm 1, see Lemma B.3 for Algorithm 2, and see Lemma B.4 for Algorithm 3.

First of all, we start with the auxiliary result.

**Lemma B.1.** *Let function $f$ satisfy Assumption 1.2 ($(L_0, L_1)$-smoothness) and Assumption 1.5 (convexity, $\mu = 0$), then for $x, y \in \mathbb{R}^d$ such that $\|y - x\| \leq \frac{1}{L_1}$ we have:*

$$\frac{\|\nabla f(y) - \nabla f(x)\|^2}{2 (L_0 + L_1 \|\nabla f(x)\|)} \leq f(x) - f(y) - \langle \nabla f(y), x - y \rangle. \tag{10}$$

*Proof.* The proof of this statement is based on the results from Nesterov (2018); Gorbunov et al. (2024).

Let us define the following function $\varphi_a(b)$ for a given $a \in \mathbb{R}^d$:

$$\varphi_a(b) = f(b) - \langle \nabla f(a), b \rangle.$$

Then this function is differentiable and $\nabla \varphi_a(b) = \nabla f(b) - \nabla f(a)$. Moreover, for any $b, c \in \mathbb{R}^d$ such that $\|b - c\| \le \frac{1}{L_1}$ we have:

$$\|\nabla \varphi_a(c) - \nabla \varphi_a(b)\| = \|\nabla f(b) - \nabla f(c)\| \overset{①}{\le} (L_0 + L_1 \|\nabla f(c)\|) \|b - c\|, \qquad (11)$$

where in ① we applied Assumption 1.2. Next, for given $a$ and for any $b, c \in \mathbb{R}^d$ such that $\|b - c\| \le \frac{1}{L_1}$ we define function $\psi_{abc}(t) : \mathbb{R} \to \mathbb{R}$ as

$$\psi_{abc}(t) = \varphi_a(c + t(b - c)).$$

Then, by definition of $\psi_{abc}$, we have $\varphi_a(c) = \psi_{abc}(0)$, $\varphi_a(b) = \psi_{abc}(1)$ and $\psi'_{abc} = \langle \nabla \varphi_a(c + t(b - c)), b - c \rangle$. Therefore, using the Newton-Leibniz formula, we have:

$$\varphi_a(b) - \varphi_a(c) = \psi_{abc}(1) - \psi_{abc}(0) = \int_0^1 \psi'_{abc} dt = \int_0^1 \langle \nabla \varphi_a(c + t(b - c)), b - c \rangle \, dt$$

$$= \langle \nabla \varphi_a(c), b - c \rangle + \int_0^1 \langle \nabla \varphi_a(c + t(b - c)) - \nabla \varphi_a(c), b - c \rangle \, dt$$

$$\overset{equation \ 7}{\le} \langle \nabla \varphi_a(c), b - c \rangle + \int_0^1 \|\nabla \varphi_a(c + t(b - c)) - \nabla \varphi_a(c)\| \|b - c\| \, dt$$

$$\overset{equation \ 11}{\le} \langle \nabla \varphi_a(c), b - c \rangle + \int_0^1 (L_0 + L_1 \|\nabla f(c)\|) \|b - c\|^2 \cdot t \cdot dt$$

$$= \langle \nabla \varphi_a(c), b - c \rangle + \frac{(L_0 + L_1 \|\nabla f(c)\|)}{2} \|b - c\|^2. \qquad (12)$$

Let $b = c - \frac{1}{L_0 + L_1 \|\nabla f(c)\|} \nabla \varphi_a(c)$ and assume that $\|a - c\| \le \frac{1}{L_1}$, then we have

$$\|b - c\| = \frac{\|\nabla \varphi_a(c)\|}{L_0 + L_1 \|\nabla f(c)\|} = \frac{\|\nabla f(c) - \nabla f(a)\|}{L_0 + L_1 \|\nabla f(c)\|} \overset{equation \ 2}{\le} \|c - a\| \le \frac{1}{L_1},$$

meaning that for this choice of $c$ and $b$ we can apply equation 12 and get:

$$\varphi_a(b) - \varphi_a(c) \le -\frac{\|\nabla \varphi_a(c)\|^2}{L_0 + L_1 \|\nabla f(c)\|} + \frac{\|\nabla \varphi_a(c)\|^2}{2 (L_0 + L_1 \|\nabla f(c)\|)} = -\frac{\|\nabla \varphi_a(c)\|^2}{2 (L_0 + L_1 \|\nabla f(c)\|)}.$$

Using the fact that $a$ is an optimum for $\varphi_a(c)$ (since $\nabla \varphi_a(a) = 0$) and by definition of $\varphi_a(c)$ we obtain the following inequality:

$$f(a) - \langle \nabla f(a), a \rangle \le f(c) - \langle \nabla f(a), c \rangle - \frac{\|\nabla f(c) - \nabla f(a)\|^2}{2 (L_0 + L_1 \|\nabla f(c)\|)}.$$

Using the fact that this inequality is satisfied for any $a, c \in \mathbb{R}^d$ such that $\|a - c\| \le \frac{1}{L_1}$, we take $a = y$ and $c = x$ and we get the original statement of the Lemma:

$$\frac{\|\nabla f(y) - \nabla f(x)\|^2}{2 (L_0 + L_1 \|\nabla f(x)\|)} \le f(x) - f(y) - \langle \nabla f(y), x - y \rangle.$$

$\qquad\qquad\qquad\qquad\qquad\qquad\qquad\qquad\qquad\qquad\qquad\qquad\qquad\qquad\qquad\qquad\qquad\quad \square$

We are now ready to present the proofs of the gradient norm monotonicity along the trajectories of the considered first-order methods.

**Lemma B.2.** *Let function $f$ satisfy Assumption 1.2 $((L_0, L_1)$-smoothness) and Assumption 1.5 (convexity, $\mu = 0$), then for all $k \ge 0$ Algorithm 1 with $\eta_k = (L_0 + L_1 \|\nabla f(x^k)\|)^{-1}$ satisfies*

$$\left\|\nabla f(x^{k+1})\right\| \le \left\|\nabla f(x^k)\right\|.$$

*Proof.* We note that for GD with $\eta_k = (L_0 + L_1 \|\nabla f(x^k)\|)^{-1}$ iterates $x^k$ and $x^{k+1}$ satisfy

$$\|x^k - x^{k+1}\| = \frac{\|\nabla f(x^k)\|}{L_0 + L_1 \|\nabla f(x^k)\|} \le \frac{1}{L_1},$$

meaning that one can apply Lemma B.1 for these points. Introducing for convenience the new notation $\omega_k = L_0 + L_1 \|\nabla f(x)\|$ and summing equation 10 with $x = x^k, y = x^{k+1}$ and $x = x^{k+1}, y = x^k$, we get the following inequality:

$$\left(\frac{1}{2\omega_k} + \frac{1}{2\omega_{k+1}}\right) \left\|\nabla f(x^{k+1}) - \nabla f(x^k)\right\|^2 \le \left\langle\nabla f(x^{k+1}) - \nabla f(x^k), x^{k+1} - x^k\right\rangle$$
$$= -\eta_k \left\langle\nabla f(x^{k+1}) - \nabla f(x^k), \nabla f(x^k)\right\rangle.$$

Multiplying both sides by $2\omega_k$, we obtain

$$\left(1 + \frac{\omega_k}{\omega_{k+1}}\right)\left(\left\|\nabla f(x^{k+1})\right\|^2 - 2\left\langle\nabla f(x^{k+1}), \nabla f(x^k)\right\rangle + \left\|\nabla f(x^k)\right\|^2\right) \le$$
$$\le -2\omega_k\eta_k\left\langle\nabla f(x^{k+1}) - \nabla f(x^k), \nabla f(x^k)\right\rangle,$$

which is equivalent to

$$\left(1 + \frac{\omega_k}{\omega_{k+1}}\right)\left\|\nabla f(x^{k+1})\right\|^2 \le \left(1 + \frac{\omega_k}{\omega_{k+1}}\right)\left\|\nabla f(x^k)\right\|^2$$
$$+ 2\left(1 + \frac{\omega_k}{\omega_{k+1}} - \omega_k\eta_k\right)\left\langle\nabla f(x^{k+1}) - \nabla f(x^k), \nabla f(x^k)\right\rangle$$
$$= \left(1 + \frac{\omega_k}{\omega_{k+1}}\right)\left\|\nabla f(x^k)\right\|^2$$
$$- \frac{2}{\eta_k}\left(1 + \frac{\omega_k}{\omega_{k+1}} - \omega_k\eta_k\right)\left\langle\nabla f(x^{k+1}) - \nabla f(x^k), x^{k+1} - x^k\right\rangle$$
$$\overset{\text{①}}{=} \left(1 + \frac{\omega_k}{\omega_{k+1}}\right)\left\|\nabla f(x^k)\right\|^2$$
$$- \frac{2\omega_k}{\omega_{k+1}\eta_k}\left\langle\nabla f(x^{k+1}) - \nabla f(x^k), x^{k+1} - x^k\right\rangle$$
$$\overset{\text{②}}{\le} \left(1 + \frac{\omega_k}{\omega_{k+1}}\right)\left\|\nabla f(x^k)\right\|^2,$$

where in ① we used $\eta_k = \frac{1}{\omega_k}$; and in ② we used $\eta_k, \omega_k, \omega_{k+1} \ge 0$ and convexity of function $f$. Hence, we obtain the original statement of the Lemma:

$$\left\|\nabla f(x^{k+1})\right\| \le \left\|\nabla f(x^k)\right\|.$$

$\square$

Next, we provide a similar result for Algorithm 2.

**Lemma B.3.** *Let function $f$ satisfy Assumption 1.2 ($(L_0, L_1)$-smoothness) and Assumption 1.5 (convexity, $\mu = 0$), then for all $k \ge 0$ Algorithm 2 with $\eta_k = \eta \le \frac{c}{L_0 + cL_1}$, where $\|\nabla f(x^k)\| \ge c$, satisfies*

$$\left\|\nabla f(x^{k+1})\right\| \le \left\|\nabla f(x^k)\right\|.$$

*Proof.* We note that for NGD with $\eta_k = \eta \le \frac{c}{L_0 + cL_1}$ iterates $x^k$ and $x^{k+1}$ satisfy

$$\|x^k - x^{k+1}\| = \eta \le \frac{1}{L_1},$$

meaning that one can apply Lemma B.1 for these points. Introducing for convenience the new notation $\omega_k = L_0 + L_1 \|\nabla f(x)\|$ and summing equation 10 with $x = x^k, y = x^{k+1}$ and $x = x^{k+1}, y = x^k$, we get the following inequality:

$$\left(\frac{1}{2\omega_k} + \frac{1}{2\omega_{k+1}}\right)\left\|\nabla f(x^{k+1}) - \nabla f(x^k)\right\|^2 \le \left\langle\nabla f(x^{k+1}) - \nabla f(x^k), x^{k+1} - x^k\right\rangle$$
$$= -\frac{\eta_k}{\|\nabla f(x^k)\|}\left\langle\nabla f(x^{k+1}) - \nabla f(x^k), \nabla f(x^k)\right\rangle.$$

Multiplying both sides by $2\omega_k$, we obtain

$$\left(1 + \frac{\omega_k}{\omega_{k+1}}\right)\left(\left\|\nabla f(x^{k+1})\right\|^2 - 2\left\langle\nabla f(x^{k+1}), \nabla f(x^k)\right\rangle + \left\|\nabla f(x^k)\right\|^2\right) \leq$$

$$\leq -\frac{2\omega_k\eta_k}{\|\nabla f(x^k)\|}\left\langle\nabla f(x^{k+1}) - \nabla f(x^k), \nabla f(x^k)\right\rangle,$$

which is equivalent to

$$\left(1 + \frac{\omega_k}{\omega_{k+1}}\right)\left\|\nabla f(x^{k+1})\right\|^2 \leq \left(1 + \frac{\omega_k}{\omega_{k+1}}\right)\left\|\nabla f(x^k)\right\|^2$$

$$+ 2\left(1 + \frac{\omega_k}{\omega_{k+1}} - \frac{\omega_k\eta_k}{\|\nabla f(x^k)\|}\right)\left\langle\nabla f(x^{k+1}) - \nabla f(x^k), \nabla f(x^k)\right\rangle$$

$$= \left(1 + \frac{\omega_k}{\omega_{k+1}}\right)\left\|\nabla f(x^k)\right\|^2$$

$$- \frac{2\|\nabla f(x^k)\|}{\eta_k}\left(1 + \frac{\omega_k}{\omega_{k+1}} - \frac{\omega_k\eta_k}{\|\nabla f(x^k)\|}\right)\left\langle\nabla f(x^{k+1}) - \nabla f(x^k), x^{k+1} - x^k\right\rangle$$

$$\overset{\text{\textcircled{1}}}{\leq} \left(1 + \frac{\omega_k}{\omega_{k+1}}\right)\left\|\nabla f(x^k)\right\|^2$$

$$- \frac{2\|\nabla f(x^k)\|}{\eta_k}\left(1 + \frac{\omega_k}{\omega_{k+1}} - \frac{\omega_k c}{\|\nabla f(x^k)\|(L_0 + L_1 c)}\right)\left\langle\nabla f(x^{k+1}) - \nabla f(x^k), x^{k+1} - x^k\right\rangle$$

$$\overset{\text{\textcircled{2}}}{\leq} \left(1 + \frac{\omega_k}{\omega_{k+1}}\right)\left\|\nabla f(x^k)\right\|^2 - \frac{2\omega_k\|\nabla f(x^k)\|}{\omega_{k+1}\eta_k}\left\langle\nabla f(x^{k+1}) - \nabla f(x^k), x^{k+1} - x^k\right\rangle$$

$$\overset{\text{\textcircled{3}}}{\leq} \left(1 + \frac{\omega_k}{\omega_{k+1}}\right)\left\|\nabla f(x^k)\right\|^2,$$

where in \textcircled{1} we used $\eta_k \leq \frac{c}{L_0 + L_1 c}$, in \textcircled{2} we used $\|\nabla f(x^k)\| \geq c$ implying $\frac{c}{L_0 + L_1 c} \leq \frac{\|\nabla f(x^k)\|}{\omega_k}$, and in \textcircled{3} we used $\|\nabla f(x^k)\|, \eta_k, \omega_k, \omega_{k+1} \geq 0$ and convexity of function $f$. Hence, we obtain the original statement of the Lemma:

$$\left\|\nabla f(x^{k+1})\right\| \leq \left\|\nabla f(x^k)\right\|.$$

$\square$

Finally, we present a similar result for Algorithm 3 that can be viewed as a combination of the previous two.

**Lemma B.4.** *Let function $f$ satisfy Assumption 1.2 ($(L_0, L_1)$-smoothness) and Assumption 1.5 (convexity, $\mu = 0$), then for all $k \geq 0$ Algorithm 3 with step size $\eta_k = (L_0 + L_1 \max\{\|\nabla f(x^k)\|, c\})^{-1}$ satisfies*

$$\left\|\nabla f(x^{k+1})\right\| \leq \left\|\nabla f(x^k)\right\|.$$

*Proof.* We note that for Clip-GD with $\eta_k = (L_0 + L_1 \max\{\|\nabla f(x^k)\|, c\})^{-1}$ iterates $x^k$ and $x^{k+1}$ satisfy

$$\|x^k - x^{k+1}\| = \frac{\max\{\|\nabla f(x^k)\|, c\}}{L_0 + L_1 \max\{\|\nabla f(x^k)\|, c\}} \leq \frac{1}{L_1},$$

meaning that one can apply Lemma B.1 for these points. Introducing for convenience the new notation $\omega_k = L_0 + L_1\|\nabla f(x)\|$ and summing equation 10 with $x = x^k, y = x^{k+1}$ and $x = x^{k+1}, y = x^k$, we get the following inequality:

$$\left(\frac{1}{2\omega_k} + \frac{1}{2\omega_{k+1}}\right)\left\|\nabla f(x^{k+1}) - \nabla f(x^k)\right\|^2 \leq$$

$$\leq \left\langle\nabla f(x^{k+1}) - \nabla f(x^k), x^{k+1} - x^k\right\rangle$$

$$= -\eta_k \cdot \underbrace{\min\left\{1, \frac{c}{\|\nabla f(x^k)\|}\right\}}_{\lambda_k}\left\langle\nabla f(x^{k+1}) - \nabla f(x^k), \nabla f(x^k)\right\rangle.$$

Multiplying both sides by $2\omega_k$, we obtain

$$\left(1 + \frac{\omega_k}{\omega_{k+1}}\right) \left(\left\|\nabla f(x^{k+1})\right\|^2 - 2\left\langle\nabla f(x^{k+1}), \nabla f(x^k)\right\rangle + \left\|\nabla f(x^k)\right\|^2\right)$$
$$\leq -2\omega_k\eta_k\lambda_k \left\langle\nabla f(x^{k+1}) - \nabla f(x^k), \nabla f(x^k)\right\rangle. \tag{13}$$

Consider two cases: $\lambda_k = 1$ or $\lambda_k = \frac{c}{\|\nabla f(x^k)\|}$. If $\lambda_k = 1$, then $c \geq \left\|\nabla f(x^k)\right\|$. Then equation 13 is equivalent to the following:

$$\left(1 + \frac{\omega_k}{\omega_{k+1}}\right) \left\|\nabla f(x^{k+1})\right\|^2 \leq \left(1 + \frac{\omega_k}{\omega_{k+1}}\right) \left\|\nabla f(x^k)\right\|^2$$
$$+ 2\left(1 + \frac{\omega_k}{\omega_{k+1}} - \omega_k\eta_k\right) \left\langle\nabla f(x^{k+1}) - \nabla f(x^k), \nabla f(x^k)\right\rangle$$
$$= \left(1 + \frac{\omega_k}{\omega_{k+1}}\right) \left\|\nabla f(x^k)\right\|^2$$
$$- \frac{2}{\eta_k}\left(1 + \frac{\omega_k}{\omega_{k+1}} - \omega_k\eta_k\right) \left\langle\nabla f(x^{k+1}) - \nabla f(x^k), x^{k+1} - x^k\right\rangle$$
$$\overset{①}{\leq} \left(1 + \frac{\omega_k}{\omega_{k+1}}\right) \left\|\nabla f(x^k)\right\|^2$$
$$- \frac{2}{\eta_k}\left(1 + \frac{\omega_k}{\omega_{k+1}} - \frac{\omega_k}{L_0 + L_1 c}\right) \left\langle\nabla f(x^{k+1}) - \nabla f(x^k), x^{k+1} - x^k\right\rangle$$
$$\overset{②}{\leq} \left(1 + \frac{\omega_k}{\omega_{k+1}}\right) \left\|\nabla f(x^k)\right\|^2 - \frac{2\omega_k}{\omega_{k+1}\eta_k} \left\langle\nabla f(x^{k+1}) - \nabla f(x^k), x^{k+1} - x^k\right\rangle$$
$$\overset{③}{\leq} \left(1 + \frac{\omega_k}{\omega_{k+1}}\right) \left\|\nabla f(x^k)\right\|^2,$$

where in ① we used $\eta_k \leq \frac{1}{L_0 + L_1 c}$, in ② we used $\frac{1}{L_0 + L_1 c} \leq \frac{1}{\omega_k}$, and in ③ we used $\eta_k, \omega_k, \omega_{k+1} \geq 0$ and convexity of function $f$.

Next, we consider the case when $\lambda_k = \frac{c}{\|\nabla f(x^k)\|}$, implying $c \leq \left\|\nabla f(x^k)\right\|$. Then, equation 13 is equivalent to the following:

$$\left(1 + \frac{\omega_k}{\omega_{k+1}}\right) \left\|\nabla f(x^{k+1})\right\|^2 \leq \left(1 + \frac{\omega_k}{\omega_{k+1}}\right) \left\|\nabla f(x^k)\right\|^2$$
$$+ 2\left(1 + \frac{\omega_k}{\omega_{k+1}} - \frac{\omega_k\eta_k c}{\|\nabla f(x^k)\|}\right) \left\langle\nabla f(x^{k+1}) - \nabla f(x^k), \nabla f(x^k)\right\rangle$$
$$= \left(1 + \frac{\omega_k}{\omega_{k+1}}\right) \left\|\nabla f(x^k)\right\|^2$$
$$- \frac{2\|\nabla f(x^k)\|}{\eta_k c}\left(1 + \frac{\omega_k}{\omega_{k+1}} - \frac{\omega_k\eta_k c}{\|\nabla f(x^k)\|}\right) \left\langle\nabla f(x^{k+1}) - \nabla f(x^k), x^{k+1} - x^k\right\rangle$$
$$\overset{①}{\leq} \left(1 + \frac{\omega_k}{\omega_{k+1}}\right) \left\|\nabla f(x^k)\right\|^2$$
$$- \frac{2\|\nabla f(x^k)\|}{\eta_k c}\left(1 + \frac{\omega_k}{\omega_{k+1}} - \frac{\omega_k}{(L_0 + L_1 c)}\right) \left\langle\nabla f(x^{k+1}) - \nabla f(x^k), x^{k+1} - x^k\right\rangle$$
$$\overset{②}{\leq} \left(1 + \frac{\omega_k}{\omega_{k+1}}\right) \left\|\nabla f(x^k)\right\|^2 - \frac{2\omega_k\left\|\nabla f(x^k)\right\|}{\omega_{k+1}\eta_k c} \left\langle\nabla f(x^{k+1}) - \nabla f(x^k), x^{k+1} - x^k\right\rangle$$
$$\overset{③}{\leq} \left(1 + \frac{\omega_k}{\omega_{k+1}}\right) \left\|\nabla f(x^k)\right\|^2,$$

where in ① we used $\eta_k \leq \frac{1}{L_0 + L_1 c}$ and $c \leq \left\|\nabla f(x^k)\right\|$, in ② we used $\frac{c}{L_0 + L_1 c} \leq \frac{1}{\omega_k}$, and in ③ we used $\left\|\nabla f(x^k)\right\|, \eta_k, \omega_k, \omega_{k+1} \geq 0$ and convexity of function $f$.

That is, in both cases, we obtain the original statement of the Lemma:

$$\left\|\nabla f(x^{k+1})\right\| \leq \left\|\nabla f(x^k)\right\|.$$

$\square$

## C  MISSING PROOFS FOR FULL-GRADIENT ALGORITHMS

In this section, we give missing proofs from the main part of the paper. In particular, see Subsection C.1 for the proof of convergence results for Algorithm 1, see Subsection C.2 for Algorithm 2, and see Subsection C.3 for Algorithm 3.

### C.1  PROOF OF THEOREM 3.1

Using Assumption 1.2, we derive

$$
\begin{aligned}
f(x^{k+1}) - f(x^k) &= f(x^k - \eta_k \nabla f(x^k)) - f(x^k) \\
&\overset{equation\ 8}{\leq} -\eta_k \left\langle \nabla f(x^k), \nabla f(x^k) \right\rangle + \eta_k^2 \frac{L_0 + L_1 \left\|\nabla f(x^k)\right\|}{2} \left\|\nabla f(x^k)\right\|^2 \\
&\overset{①}{\leq} -\eta_k \left\|\nabla f(x^k)\right\|^2 + \frac{\eta_k}{2} \left\|\nabla f(x^k)\right\|^2 \\
&= -\frac{\eta_k}{2} \left\|\nabla f(x^k)\right\|^2,
\end{aligned}
\tag{14}
$$

where in ① we used $\eta_k \leq \frac{1}{L_0 + L_1 \|\nabla f(x^k)\|}$. Next, let us consider two cases.

$\boxed{\text{The case of } \left\|\nabla f(x^k)\right\| \geq \frac{L_0}{L_1}.}$ Taking $\eta_k = \frac{1}{L_0 + L_1 \|\nabla f(x^k)\|}$ and using the convexity assumption of the function (see Assumption 1.5, $\mu = 0$), we have the following:

$$
f(x^k) - f^* \leq \left\langle \nabla f(x^k), x^k - x^* \right\rangle \overset{equation\ 7}{\leq} \left\|\nabla f(x^k)\right\| \left\|x^k - x^*\right\| \overset{①}{\leq} \left\|\nabla f(x^k)\right\| \underbrace{\left\|x^0 - x^*\right\|}_{R}
$$

$$
= \frac{\eta_k}{\eta_k} \left\|\nabla f(x^k)\right\| R = \eta_k (L_0 + L_1 \left\|\nabla f(x^k)\right\|) \left\|\nabla f(x^k)\right\| R \leq 2\eta_k L_1 \left\|\nabla f(x^k)\right\|^2 R,
$$

where ① follows from $\|x^k - x^*\| \leq \|x^0 - x^*\|$ (Gorbunov et al., 2024, proof of Theorem 3.3). The above inequality implies

$$
\eta_k \geq \frac{f(x^k) - f^*}{2L_1 R \left\|\nabla f(x^k)\right\|^2}.
\tag{15}
$$

Plugging equation 15 into equation 14, we obtain

$$
f(x^{k+1}) - f(x^k) \leq -\eta_k \left\|\nabla f(x^k)\right\|^2 \leq \frac{1}{4L_1 R}(f(x^k) - f^*),
$$

which is equivalent to

$$
f(x^{k+1}) - f^* \leq \left(1 - \frac{1}{4L_1 R}\right)(f(x^k) - f^*).
\tag{16}
$$

Moreover, Lemma B.2 implies that for all $t = 0, \ldots, k$ a similar inequality holds. We denote $T := \min\left\{k \in \{0, 1, 2, ..., N-1\} \mid \left\|\nabla f(x^k)\right\| < \frac{L_0}{L_1} \text{ and } \left\|\nabla f(x^{k-1})\right\| \geq \frac{L_0}{L_1}\right\}$ as the first index $k$ such that $\left\|\nabla f(x^k)\right\| < \frac{L_0}{L_1}$ (note that $T = 0$ is possible). Then, for the first $T$ iterations, we have exponential objective decrease:

$$
f(x^T) - f^* \leq \left(1 - \frac{1}{4L_1 R}\right)^T (f(x^0) - f^*),
\tag{17}
$$

which follows from unrolling equation 16.

The case of $\left\|\nabla f(x^k)\right\| < \frac{L_0}{L_1}$. Taking $\eta_k = \frac{1}{L_0 + L_1 \|\nabla f(x^k)\|}$ and using the convexity assumption of the function (see Assumption 1.5, $\mu = 0$), we have the following:

$$f(x^k) - f^* \le \left\langle \nabla f(x^k), x^k - x^* \right\rangle \overset{equation\ 7}{\le} \left\|\nabla f(x^k)\right\| \left\|x^k - x^*\right\| \le \left\|\nabla f(x^k)\right\| \underbrace{\left\|x^0 - x^*\right\|}_{R}$$

$$(18)$$

$$= \frac{\eta_k}{\eta_k} \left\|\nabla f(x^k)\right\| R = \eta_k (L_0 + L_1 \left\|\nabla f(x^k)\right\|) \left\|\nabla f(x^k)\right\| R < 2\eta_k L_0 \left\|\nabla f(x^k)\right\| R.$$

The above inequality implies

$$\eta_k > \frac{f(x^k) - f^*}{2 L_0 R \left\|\nabla f(x^k)\right\|}. \tag{19}$$

Then, plugging equation 19 into equation 14 and using the notation $F_k = f(x^k) - f^*$, we obtain:

$$F_{k+1} < F_k - \frac{\left\|\nabla f(x^k)\right\|}{4 L_0 R} F_k \overset{equation\ 18}{\le} F_k - \frac{1}{4 L_0 R^2} F_k^2,$$

which is equivalent to

$$\frac{1}{4 L_0 R^2} F_k^2 < F_k - F_{k+1}.$$

Next, we divide both sides by $F_{k+1} F_k$

$$\frac{1}{4 L_0 R^2} \cdot \frac{F_k}{F_{k+1}} < \frac{1}{F_{k+1}} - \frac{1}{F_k}$$

and use that $F_{k+1} \le F_k$ due to equation 14:

$$\frac{1}{4 L_0 R^2} < \frac{1}{F_{k+1}} - \frac{1}{F_k}.$$

Summing up the above inequality for $k = T, T+1, ..., N$, we get

$$\frac{N-T}{4 L_0 R^2} = \sum_{k=T}^{N-1} \frac{1}{4 L_0 R^2} < \sum_{k=T}^{N-1} \left( \frac{1}{F_{k+1}} - \frac{1}{F_k} \right) = \frac{1}{F_N} - \frac{1}{F_T} < \frac{1}{F_N},$$

which is equivalent to

$$f(x^N) - f^* < \frac{4 L_0 R^2}{N - T}. \tag{20}$$

Finally, combining inequalities equation 17 and equation 20 and taking into account that $F_N \le F_T$, we obtain the convergence rate of Algorithm 1 in the convex case:

$$f(x^N) - f^* = \mathcal{O}\left( \min\left\{ \frac{L_0 R^2}{N - T}, \left(1 - \frac{1}{L_1 R}\right)^T F_0 \right\} \right),$$

where $T := \min\left\{ k \in \{0, 1, 2, ..., N-1\} \mid \left\|\nabla f(x^k)\right\| < \frac{L_0}{L_1} \text{ and } \left\|\nabla f(x^{k-1})\right\| \ge \frac{L_0}{L_1} \right\}$.

## C.2 PROOF OF THEOREM 3.3

Using Assumption 1.2, we derive

$$f(x^{k+1}) - f(x^k) = f\left( x^k - \eta_k \frac{\nabla f(x^k)}{\|\nabla f(x^k)\|} \right) - f(x^k)$$

$$\overset{equation\ 8}{\le} -\frac{\eta_k}{\|\nabla f(x^k)\|} \left\langle \nabla f(x^k), \nabla f(x^k) \right\rangle + \eta_k^2 \frac{L_0 + L_1 \left\|\nabla f(x^k)\right\|}{2 \|\nabla f(x^k)\|^2} \left\|\nabla f(x^k)\right\|^2$$

$$\overset{①}{\le} -\eta_k \left\|\nabla f(x^k)\right\| + \frac{\eta_k}{2} \left\|\nabla f(x^k)\right\|$$

$$= -\frac{\eta_k}{2} \left\|\nabla f(x^k)\right\|, \tag{21}$$

where in ① we used $\eta_k = \eta \leq \frac{c}{L_0+L_1c} \leq \frac{\|\nabla f(x^k)\|}{L_0+L_1\|\nabla f(x^k)\|}$ since $\|\nabla f(x^k)\| \geq c$ for all $k = 0, 1, \ldots, N-1$ and function $\varphi(u) = \frac{u}{L_0+L_1u}$ is increasing function in $u \geq 0$.

Next, we us the convexity assumption of the function (see Assumption 1.5, $\mu = 0$):

$$f(x^k) - f^* \leq \langle \nabla f(x^k), x^k - x^* \rangle \overset{equation\ 7}{\leq} \|\nabla f(x^k)\| \, \|x^k - x^*\| \overset{①}{\leq} \|\nabla f(x^k)\| \, \underbrace{\|x^0 - x^*\|}_{R},$$

$$(22)$$

where ① follows from $\|x^k - x^*\| \leq \|x^0 - x^*\|$:

$$\|x^k - x^*\|^2 = \|x^{k-1} - x^*\|^2 - \frac{2\eta_k}{\|\nabla f(x^k)\|} \langle \nabla f(x^k), x^k - x^* \rangle + \eta_k^2$$

$$\overset{equation\ 3}{\leq} \|x^{k-1} - x^*\|^2 - \frac{2\eta(f(x^k) - f^*)}{\|\nabla f(x^k)\|} + \eta^2$$

$$= \|x^{k-1} - x^*\|^2 - \eta \left( \frac{2(f(x^k) - f^*)}{\|\nabla f(x^k)\|} - \eta \right)$$

$$\leq \|x^{k-1} - x^*\|^2,$$

where in the last step, we use

$$\frac{\eta\|\nabla f(x^k)\|}{2} \leq \frac{c\|\nabla f(x^k)\|}{2(L_0+L_1c)} \leq \frac{\|\nabla f(x^k)\|^2}{2(L_0 + L_1 \|\nabla f(x^k)\|)} \overset{equation\ 10}{\leq} f(x^k) - f^*.$$

Next, inequality equation 22 gives

$$\|\nabla f(x^k)\| \geq \frac{f(x^k) - f^*}{R}. \tag{23}$$

Then, plugging equation 23 into equation 21, we obtain:

$$f(x^{k+1}) - f(x^k) \leq -\frac{\eta}{2} \|\nabla f(x^k)\| \leq -\frac{\eta}{2R}(f(x^k) - f^*),$$

which is equivalent to

$$f(x^{k+1}) - f^* \leq \left(1 - \frac{\eta}{2R}\right) (f(x^k) - f^*).$$

Unrolling the above recurrence, we derive the linear convergence for NGD with step size $\eta_k = \eta \leq \frac{c}{L_0+L_1c}$:

$$f(x^N) - f^* \leq \left(1 - \frac{\eta}{2R}\right)^N (f(x^0) - f^*).$$

### C.3  PROOF OF THEOREM 3.5

Since $\lambda_k = \min\left\{1, \frac{c}{\|\nabla f(x^k)\|}\right\}$, we there are only two possible cases for $\lambda_k$: either $\lambda_k = 1$ or $\lambda_k = \frac{c}{\|\nabla f(x^k)\|}$.

i) Consider the case of $\lambda_k = \frac{c}{\|\nabla f(x^k)\|}$, i.e., $c \leq \|\nabla f(x^k)\|$. Using Assumption 1.2, we derive

$$
\begin{aligned}
f(x^{k+1}) - f(x^k) &= f(x^k - \eta_k \lambda_k \nabla f(x^k)) - f(x^k) \\
&= f\left(x^k - \eta_k \frac{c}{\|\nabla f(x^k)\|} \nabla f(x^k)\right) - f(x^k) \\
&\overset{equation\ 8}{\leq} -\eta_k \frac{c}{\|\nabla f(x^k)\|} \left\langle \nabla f(x^k), \nabla f(x^k) \right\rangle \\
&\quad + \eta_k^2 \frac{c^2}{\|\nabla f(x^k)\|^2} \frac{L_0 + L_1 \|\nabla f(x^k)\|}{2} \|\nabla f(x^k)\|^2 \\
&= -\eta_k c \|\nabla f(x^k)\| + \eta_k^2 c^2 \frac{L_0 + L_1 \|\nabla f(x^k)\|}{2} \\
&\overset{①}{\leq} -\eta_k c \|\nabla f(x^k)\| + \frac{\eta_k c}{2} \|\nabla f(x^k)\| \\
&= -\frac{\eta_k c}{2} \|\nabla f(x^k)\|,
\end{aligned}
\tag{24}
$$

where in ① we used $\eta_k \leq \frac{\|\nabla f(x^k)\|}{c(L_0 + L_1 \|\nabla f(x^k)\|)}$, which follows from $c \leq \|\nabla f(x^k)\|$:

$$
\frac{\|\nabla f(x^k)\|}{c(L_0 + L_1 \|\nabla f(x^k)\|)} = \frac{1}{L_0 \frac{c}{\|\nabla f(x^k)\|} + L_1 c} \geq \frac{1}{L_0 + L_1 c} = \eta = \eta_k.
$$

Next, using the convexity assumption of the function (see Assumption 1.5, $\mu = 0$), we get

$$
\begin{aligned}
f(x^k) - f^* \leq \left\langle \nabla f(x^k), x^k - x^* \right\rangle &\overset{equation\ 7}{\leq} \|\nabla f(x^k)\| \|x^k - x^*\| \\
&\overset{①}{\leq} \|\nabla f(x^k)\| \underbrace{\|x^0 - x^*\|}_{R},
\end{aligned}
\tag{25}
$$

where ① follows from $\|x^k - x^*\| \leq \|x^0 - x^*\|$:

$$
\begin{aligned}
\|x^k - x^*\|^2 &= \|x^{k-1} - x^*\|^2 - \frac{2c\eta_k}{\|\nabla f(x^k)\|} \left\langle \nabla f(x^k), x^k - x^* \right\rangle + c^2 \eta_k^2 \\
&\overset{equation\ 3}{\leq} \|x^{k-1} - x^*\|^2 - \frac{2c\eta(f(x^k) - f^*)}{\|\nabla f(x^k)\|} + c^2 \eta^2 \\
&= \|x^{k-1} - x^*\|^2 - c\eta \left(\frac{2(f(x^k) - f^*)}{\|\nabla f(x^k)\|} - c\eta\right) \\
&\leq \|x^{k-1} - x^*\|^2,
\end{aligned}
$$

where in the last step, we use

$$
\frac{c\eta \|\nabla f(x^k)\|}{2} \leq \frac{c\|\nabla f(x^k)\|}{2(L_0 + L_1 c)} \leq \frac{\|\nabla f(x^k)\|^2}{2(L_0 + L_1 \|\nabla f(x^k)\|)} \overset{equation\ 10}{\leq} f(x^k) - f^*.
$$

Inequality equation 25 gives

$$
\|\nabla f(x^k)\| \geq \frac{f(x^k) - f^*}{R}.
\tag{26}
$$

Then, plugging equation 26 into equation 24, we obtain

$$
f(x^{k+1}) - f(x^k) \overset{equation\ 24}{\leq} -\frac{\eta c}{2} \|\nabla f(x^k)\| \leq \frac{\eta c}{2R}(f(x^k) - f^*),
$$

which is equivalent to

$$
f(x^{k+1}) - f^* \leq \left(1 - \frac{\eta c}{2R}\right) (f(x^k) - f^*).
$$

Next, we consider two possible scenarios for the convergence of the algorithm depending on the relation between $\|\nabla f(x^k)\|$, $c$ and $\frac{L_0}{L_1}$ (note that $\|\nabla f(x^k)\| \geq c$ in this case), given the monotonicity of the gradient norm (Lemma B.4).

($\mathcal{T}$) If for $k = 0, 1, 2, ..., \mathcal{T}_1 - 1$, the iterates of Clip-GD satisfy $\left\|\nabla f(x^k)\right\| \geq c \geq \frac{L_0}{L_1}$, then $\eta \geq \frac{1}{2L_1 c}$ and we have exponential objective decrease for the first $\mathcal{T}_1$ iterations:

$$f(x^{\mathcal{T}_1}) - f^* \leq \left(1 - \frac{1}{4L_1 R}\right)^{\mathcal{T}_1} \left(f(x^0) - f^*\right). \tag{27}$$

($\mathcal{K}$) If for $k = 0, 1, 2, ..., \mathcal{K}_1 - 1$, the iterates of Clip-GD satisfy $\left\|\nabla f(x^k)\right\| \geq \frac{L_0}{L_1} \geq c$ or $\frac{L_0}{L_1} \geq \left\|\nabla f(x^k)\right\| \geq c$, then $\eta \geq \frac{1}{2L_0}$ and we have exponential objective decrease for the first $\mathcal{K}_1$ iterations:

$$f(x^{\mathcal{K}_1}) - f^* \leq \left(1 - \frac{c}{4L_0 R}\right)^{\mathcal{K}_1} \left(f(x^0) - f^*\right). \tag{28}$$

ii) Consider the case of $\lambda_k = 1$, i.e., $c \geq \|\nabla f(x^k)\|$. Using Assumption 1.2, we derive

$$
\begin{aligned}
f(x^{k+1}) - f(x^k) &= f(x^k - \eta_k \lambda_k \nabla f(x^k)) - f(x^k) \\
&= f(x^k - \eta_k \nabla f(x^k)) - f(x^k) \\
&\overset{\text{equation 8}}{\leq} -\eta_k \left\langle \nabla f(x^k), \nabla f(x^k) \right\rangle + \eta_k^2 \frac{L_0 + L_1 \left\|\nabla f(x^k)\right\|}{2} \left\|\nabla f(x^k)\right\|^2 \\
&= -\eta_k \left\|\nabla f(x^k)\right\|^2 + \eta_k^2 \frac{L_0 + L_1 \left\|\nabla f(x^k)\right\|}{2} \left\|\nabla f(x^k)\right\|^2 \\
&\overset{①}{=} -\frac{1}{2(L_0 + L_1 \left\|\nabla f(x^k)\right\|)} \left\|\nabla f(x^k)\right\|^2,
\end{aligned} \tag{29}
$$

where in ① we used $\eta_k = \frac{1}{L_0 + L_1 \|\nabla f(x^k)\|}$. Using the convexity assumption of the function (see Assumption 1.5, $\mu = 0$), we get

$$
\begin{aligned}
f(x^k) - f^* &\leq \left\langle \nabla f(x^k), x^k - x^* \right\rangle \overset{\text{equation 7}}{\leq} \left\|\nabla f(x^k)\right\| \left\|x^k - x^*\right\| \\
&\overset{①}{\leq} \left\|\nabla f(x^k)\right\| \underbrace{\left\|x^0 - x^*\right\|}_{R},
\end{aligned} \tag{30}
$$

where ① follows from $\|x^k - x^*\| \leq \|x^0 - x^*\|$:

$$
\begin{aligned}
\|x^k - x^*\|^2 &= \|x^{k-1} - x^*\|^2 - 2\eta_k \langle \nabla f(x^k), x^k - x^* \rangle + \eta_k^2 \|\nabla f(x^k)\|^2 \\
&\overset{\text{equation 3}}{\leq} \|x^{k-1} - x^*\|^2 - 2\eta_k (f(x^k) - f^*) + \eta_k^2 \|\nabla f(x^k)\|^2 \\
&= \|x^{k-1} - x^*\|^2 - \eta_k \left(2(f(x^k) - f^*) - \eta_k \|\nabla f(x^k)\|^2\right) \\
&\leq \|x^{k-1} - x^*\|^2,
\end{aligned}
$$

where in the last step, we use

$$\frac{\eta_k \|\nabla f(x^k)\|}{2} \leq \frac{\left\|\nabla f(x^k)\right\|^2}{2(L_0 + L_1 \left\|\nabla f(x^k)\right\|)} \overset{\text{equation 10}}{\leq} f(x^k) - f^*.$$

Inequality equation 30, implies

$$\left\|\nabla f(x^k)\right\| \geq \frac{f(x^k) - f^*}{R}. \tag{31}$$

Next, we consider two cases: $\left\|\nabla f(x^k)\right\| \geq \frac{L_0}{L_1}$ and $\left\|\nabla f(x^k)\right\| < \frac{L_0}{L_1}$.

The case of $\left\|\nabla f(x^k)\right\| \geq \frac{L_0}{L_1}$. In this case, inequality equation 29 gives

$$f(x^{k+1}) - f(x^k) \leq -\frac{1}{4L_1} \left\|\nabla f(x^k)\right\|. \tag{32}$$

Then, plugging equation 31 into equation 32, we obtain:

$$f(x^{k+1}) - f(x^k) \le -\frac{1}{4L_1 R}(f(x^k) - f^*),$$

which is equivalent to

$$f(x^{k+1}) - f^* \le \left(1 - \frac{1}{4L_1 R}\right)(f(x^k) - f^*).$$

Since in this case we have the following relation $c \ge \left\|\nabla f(x^k)\right\| \ge \frac{L_0}{L_1}$, then for $k = \mathcal{T}_1, \mathcal{T}_1 + 1, ..., \mathcal{T}_2 - 1$ we have exponential objective decrease:

$$f(x^{\mathcal{T}_2}) - f^* \le \left(1 - \frac{1}{4L_1 R}\right)^{\mathcal{T}_2 - \mathcal{T}_1}(f(x^{\mathcal{T}_1}) - f^*)$$

$$\overset{equation\ 27}{\le} \left(1 - \frac{1}{4L_1 R}\right)^{\mathcal{T}_2}(f(x^0) - f^*). \tag{33}$$

The case of $\left\|\nabla f(x^k)\right\| < \frac{L_0}{L_1}.$ In this case, inequality equation 29 gives

$$f(x^{k+1}) - f(x^k) \le -\frac{1}{2(L_0 + L_1\left\|\nabla f(x^k)\right\|)}\left\|\nabla f(x^k)\right\|^2$$

$$< -\frac{1}{4L_0}\left\|\nabla f(x^k)\right\|^2. \tag{34}$$

Then, plugging equation 31 into equation 34 and using the notation $F_k := f(x^k) - f^*$, we obtain:

$$F_{k+1} < F_k - \frac{\left\|\nabla f(x^k)\right\|}{4L_0 R}F_k \le F_k - \frac{1}{4L_0 R^2}F_k^2,$$

which is equivalent to

$$\frac{1}{4L_0 R^2}F_k^2 < F_k - F_{k+1}.$$

Next, we divide both sides by $F_{k+1}F_k$

$$\frac{1}{4L_0 R^2} \cdot \frac{F_k}{F_{k+1}} < \frac{1}{F_{k+1}} - \frac{1}{F_k}.$$

and use $F_{k+1} \le F_k$ due to equation 34:

$$\frac{1}{4L_0 R^2} < \frac{1}{F_{k+1}} - \frac{1}{F_k}. \tag{35}$$

Then, two situations are possible: either $\frac{L_0}{L_1} > c$ or $\frac{L_0}{L_1} \le c$. We consider each of them separately.

($\mathcal{K}$) Considering the scenario $\frac{L_0}{L_1} > c > \left\|\nabla f(x^k)\right\|$ and summing up inequality equation 35 for $k = \mathcal{K}_1, \mathcal{K}_1 + 1, ..., \mathcal{K}_2 - 1$, we get

$$\frac{\mathcal{K}_2 - \mathcal{K}_1}{4L_0 R^2} = \sum_{k=\mathcal{K}_1}^{\mathcal{K}_2 - 1}\frac{1}{4L_0 R^2} < \sum_{k=\mathcal{K}_1}^{\mathcal{K}_2 - 1}\left(\frac{1}{F_{k+1}} - \frac{1}{F_k}\right) = \frac{1}{F_{\mathcal{K}_2}} - \frac{1}{F_{\mathcal{K}_1}} < \frac{1}{F_{\mathcal{K}_2}},$$

which is equivalent to

$$f(x^{\mathcal{K}_2}) - f^* < \frac{4L_0 R^2}{\mathcal{K}_2 - \mathcal{K}_1}. \tag{36}$$

($\mathcal{T}$) Considering the scenario $c \ge \frac{L_0}{L_1} > \left\|\nabla f(x^k)\right\|$ and summing up inequality equation 35 for $k = \mathcal{T}_2, \mathcal{T}_2 + 1, ..., \mathcal{T}_3 - 1$, we get

$$\frac{\mathcal{T}_3 - \mathcal{T}_2}{4L_0 R^2} = \sum_{k=\mathcal{T}_2}^{\mathcal{T}_3 - 1}\frac{1}{4L_0 R^2} < \sum_{k=\mathcal{T}_2}^{\mathcal{T}_3 - 1}\left(\frac{1}{F_{k+1}} - \frac{1}{F_k}\right) = \frac{1}{F_{\mathcal{T}_3}} - \frac{1}{F_{\mathcal{T}_2}} < \frac{1}{F_{\mathcal{T}_3}},$$

which is equivalent to

$$f(x^{\mathcal{T}_3}) - f^* < \frac{4L_0 R^2}{\mathcal{T}_3 - \mathcal{T}_2}. \tag{37}$$

Finally, combining equation 27, equation 28, equation 33, equation 36, equation 37, and taking into account that $F_{k+1} \leq F_k$, we obtain the convergence rate for Algorithm 3.

- If $\boxed{c \geq \frac{L_0}{L_1},}$ then for $\mathcal{T}_3 = N$ being the total number of iterations of Algorithm 3 the iterates satisfy (see equation 27, equation 33 and equation 37):

$$f(x^N) - f^* = \mathcal{O}\left(\min\left\{\frac{L_0 R^2}{N - \mathcal{T}_2}, \left(1 - \frac{1}{L_1 R}\right)^{\mathcal{T}_2} F_0\right\}\right),$$

where $\mathcal{T}_2 := \min\left\{k \in \{0, 1, ..., N-1\}\mid \|\nabla f(x^k)\| < \frac{L_0}{L_1} \text{ and } \|\nabla f(x^{k-1})\| \geq \frac{L_0}{L_1}\right\}.$

- If $\boxed{c < \frac{L_0}{L_1},}$ then $\mathcal{K}_2 = N$ being the total number of iterations of Algorithm 3 the iterates satisfy (see equation 28 and equation 36):

$$f(x^N) - f^* = \mathcal{O}\left(\min\left\{\frac{L_0 R^2}{N - \mathcal{K}_1}, \left(1 - \frac{c}{L_0 R}\right)^{\mathcal{K}_1} F_0\right\}\right),$$

where $\mathcal{K}_1 :=:= \min\left\{k \in \{0, 1, ..., N-1\}\mid \|\nabla f(x^k)\| < c \text{ and } \|\nabla f(x^{k-1})\| \geq c\right\}.$

It is not difficult to see that these two scenarios can be combined into the following equivalent form:

$$f(x^N) - f^* = \mathcal{O}\left(\min\left\{\frac{L_0 R^2}{N - T}, \left(1 - \frac{c}{\max\{L_0, L_1 c\} R}\right)^T F_0\right\}\right),$$

where $T := \min\left\{k \in \{0, 1, ..., N-1\}\mid \|\nabla f(x^k)\| < \min\left\{\frac{L_0}{L_1}, c\right\} \text{ and } \|\nabla f(x^{k-1})\| \geq \min\left\{\frac{L_0}{L_1}, c\right\}\right\}.$

## D MISSING PROOFS FOR COORDINATE DESCENT TYPE METHODS

In this section, we provide missing proofs from Section 4. In particular, see Subsection D.1 for the proof of the convergence results for Algorithm 4, and see Subsection D.2 for Algorithm 5.

### D.1 PROOF OF THEOREM 4.1

Using Assumption 1.4, we derive

$$
\begin{aligned}
f(x^{k+1}) - f(x^k) &= f(x^k - \eta_k \nabla_{i_k} f(x^k) \mathbf{e}_{i_k}) - f(x^k) \\
&\overset{equation\ 9}{\leq} -\eta_k \left(\nabla_{i_k} f(x^k)\right)^2 + \eta_k^2 \frac{L_0 + L_1 |\nabla f(x^k)|}{2} \left(\nabla_{i_k} f(x^k)\right)^2 \\
&\overset{①}{\leq} -\eta_k \left(\nabla_{i_k} f(x^k)\right)^2 + \frac{\eta_k}{2} \left(\nabla_{i_k} f(x^k)\right)^2 \\
&= -\frac{\eta_k}{2} \left(\nabla_{i_k} f(x^k)\right)^2,
\end{aligned}
\tag{38}
$$

where in ① we used $\eta_k \le \frac{1}{L_0 + L_1 |\nabla_{i_k} f(x^k)|}$. Next, we take the expectation w.r.t. $i_k$ and use $\eta_k = \frac{1}{L_0 + L_1 |\nabla_{i_k} f(x^k)|}$:

$$\mathbb{E}_{i_k}[f(x^{k+1})] - f(x^k) \le -\frac{1}{2d} \sum_{i=1}^{d} \frac{|\nabla_i f(x^k)|^2}{L_0 + L_1 |\nabla_i f(x^k)|}$$

$$\le -\frac{1}{4d} \sum_{i=1}^{d} \min\left\{ \frac{|\nabla_i f(x^k)|^2}{L_0}, \frac{|\nabla_i f(x^k)|}{L_1} \right\} \qquad (39)$$

$$= -\frac{1}{4d} \left( \sum_{i \in I_k} \frac{|\nabla_i f(x^k)|}{L_1} + \sum_{i \in [d] \setminus I_k} \frac{|\nabla_i f(x^k)|^2}{L_0} \right), \qquad (40)$$

where $I_k := \left\{ i \in [d] \mid |\nabla_i f(x^k)| \ge \frac{L_0}{L_1} \right\}$. Next, we introduce the set of indices $\mathcal{K}$ as $\mathcal{K} := \left\{ k \in [N-1] \mid \sum_{i \in I_k} |\nabla_i f(x^k)|^2 > \sum_{i \in [d] \setminus I_k} |\nabla_i f(x^k)|^2 \right\}$ and consider two possible situations.

The case of $k \in \mathcal{K}$. In this case, we continue our derivation as follows:

$$\mathbb{E}_{i_k}[f(x^{k+1})] - f(x^k) \le -\frac{1}{4dL_1} \sum_{i \in I_k} |\nabla_i f(x^k)|. \qquad (41)$$

Using the convexity assumption and notation $F_k := f(x^k) - f^*$, we derive

$$F_k \le \langle \nabla f(x^k), x^k - x^* \rangle \overset{equation\ 7}{\le} \|\nabla f(x^k)\| \underbrace{\|x^k - x^*\|}_{R}$$

$$= R \sqrt{\sum_{i \in I_k} |\nabla_i f(x^k)|^2 + \sum_{i \in [d] \setminus I_k} |\nabla_i f(x^k)|^2} \le R \sqrt{2 \sum_{i \in I_k} |\nabla_i f(x^k)|^2} \le \sqrt{2} R \sum_{i \in I_k} |\nabla_i f(x^k)|$$

that implies

$$\sum_{i \in I_k} |\nabla_i f(x^k)| \ge \frac{F_k}{\sqrt{2} R}. \qquad (42)$$

Plugging equation 42 in equation 41, we obtain

$$\mathbb{E}_{i_k}[F_{k+1}] \le \left( 1 - \frac{1}{4\sqrt{2} d L_1 R} \right) F_k. \qquad (43)$$

The case of $k \notin \mathcal{K}$. In this case, we continue equation 40 as follows:

$$\mathbb{E}_{i_k}[f(x^{k+1})] - f(x^k) \le -\frac{1}{4dL_0} \sum_{i \in [d] \setminus I_k} |\nabla_i f(x^k)|^2. \qquad (44)$$

Using the convexity assumption and notation $F_k := f(x^k) - f^*$, we derive

$$F_k \le \langle \nabla f(x^k), x^k - x^* \rangle \overset{equation\ 7}{\le} \|\nabla f(x^k)\| \underbrace{\|x^k - x^*\|}_{R}$$

$$= R \sqrt{\sum_{i \in I_k} |\nabla_i f(x^k)|^2 + \sum_{i \in [d] \setminus I_k} |\nabla_i f(x^k)|^2} \le R \sqrt{2 \sum_{i \in [d] \setminus I_k} |\nabla_i f(x^k)|^2}$$

that implies

$$\sum_{i \in I_k} |\nabla_i f(x^k)|^2 \ge \frac{F_k^2}{2R^2}. \qquad (45)$$

Plugging equation 45 in equation 44, we obtain

$$\mathbb{E}_{i_k}[F_{k+1}] \leq F_k - \frac{1}{8dL_0R^2}F_k^2. \tag{46}$$

To get the final bound, let us specify the indices belonging to $\mathcal{K}$: let $\mathcal{K} := \{k_1, k_2, \ldots, k_r\}$ and $\mathcal{T} := [N-1] \setminus \mathcal{K} := \{t_1, t_2, \ldots, t_{N-r}\}$, where $0 \leq k_1 \leq k_2 \leq \ldots \leq k_r \leq N-1$ and $0 \leq t_1 \leq t_2 \leq \ldots \leq t_{N-r} \leq N-1$. Note that $\mathcal{K} \cap \mathcal{T} = \varnothing$, $\mathcal{K} \cup \mathcal{T} = [N-1]$, and $|\mathcal{K}| = r$ is random variable. There exist two possible situations: either $r > N/2$ or $r \leq N/2$. If $r > N/2$, then we use equation 43 together with $F_{k+1} \leq F_k$ following from equation 38:

$$\mathbb{E}_{i \in \mathcal{K}}[F_N] \overset{\text{equation } 38}{\leq} \mathbb{E}_{i \in \mathcal{K}}[F_{k_r+1}] \overset{\text{equation } 43}{\leq} \left(1 - \frac{1}{4\sqrt{2}dL_1R}\right) \mathbb{E}_{i \in \mathcal{K} \setminus \{k_r\}}[F_{k_r}]$$

$$\overset{\text{equation } 38}{\leq} \left(1 - \frac{1}{4\sqrt{2}dL_1R}\right) \mathbb{E}_{i \in \mathcal{K} \setminus \{k_r\}}[F_{k_{r-1}+1}] \overset{\text{equation } 43}{\leq} \left(1 - \frac{1}{4\sqrt{2}dL_1R}\right)^2 \mathbb{E}_{i \in \mathcal{K} \setminus \{k_r, k_{r-1}\}}[F_{k_{r-1}}]$$

$$\leq \ldots \leq \left(1 - \frac{1}{4\sqrt{2}dL_1R}\right)^r F_0 \overset{r > N/2}{\leq} \left(1 - \frac{1}{4\sqrt{2}dL_1R}\right)^{N/2} F_0 \mathbb{1}_{\{r > N/2\}}, \tag{47}$$

where $\mathbb{E}_{i \in \mathcal{K}}$ denotes the expectation w.r.t. all indices in set $\mathcal{K}$ and $\mathbb{1}_{\{r > N/2\}}$ is an indicator of the event $\{r > N/2\}$.

Next, we consider the situation when $r \leq N/2$. In this case, we first notice that equation 46 gives

$$\mathbb{E}_{i \in \mathcal{T}}[F_{t_{N-r}+1}] \overset{\text{equation } 46}{\leq} \mathbb{E}_{i \in \mathcal{T} \setminus \{t_{N-r}\}}[F_{t_{N-r}}] - \frac{1}{8dL_0R^2}\mathbb{E}_{i \in \mathcal{T} \setminus \{t_{N-r}\}}[F_{t_{N-r}}^2]$$

$$\overset{\text{①}}{\leq} \mathbb{E}_{i \in \mathcal{T} \setminus \{t_{N-r}\}}[F_{t_{N-r}}] - \frac{1}{8dL_0R^2}\mathbb{E}_{i \in \mathcal{T} \setminus \{t_{N-r}\}}[F_{t_{N-r}}]^2$$

where in ① we used $\mathbb{E}_{i \in \mathcal{T} \setminus \{t_{N-r}\}}[F_{t_{N-r}}]^2 \leq \mathbb{E}_{i \in \mathcal{T} \setminus \{t_{N-r}\}}[F_{t_{N-r}}^2]$. Dividing both sides by $\mathbb{E}_{i \in \mathcal{T}}[F_{t_{N-r}+1}]\mathbb{E}_{i \in \mathcal{T} \setminus \{t_{N-r}\}}[F_{t_{N-r}}]$ and rearranging the terms, we get

$$\frac{1}{8dL_0R^2}\frac{\mathbb{E}_{i \in \mathcal{T} \setminus \{t_{N-r}\}}[F_{t_{N-r}}]}{\mathbb{E}_{i \in \mathcal{T}}[F_{t_{N-r}+1}]} \leq \frac{1}{\mathbb{E}_{i \in \mathcal{T}}[F_{t_{N-r}+1}]} - \frac{1}{\mathbb{E}_{i \in \mathcal{T} \setminus \{t_{N-r}\}}[F_{t_{N-r}}]}. \tag{48}$$

In view of equation 38, we have $\mathbb{E}_{i \in \mathcal{T}}[F_{t_{N-r}+1}] \leq \mathbb{E}_{i \in \mathcal{T}}[F_{t_{N-r}}] = \mathbb{E}_{i \in \mathcal{T} \setminus \{t_{N-r}\}}[F_{t_{N-r}}]$ and $-\frac{1}{\mathbb{E}_{i \in \mathcal{T} \setminus \{t_{N-r}\}}[F_{t_{N-r}}]} \leq -\frac{1}{\mathbb{E}_{i \in \mathcal{T} \setminus \{t_{N-r}\}}[F_{t_{N-r-1}+1}]} = -\frac{1}{\mathbb{E}_{i \in \mathcal{T}}[F_{t_{N-r-1}+1}]}$. Using these inequalities in equation 48, we obtain

$$\frac{1}{8dL_0R^2} \leq \frac{1}{\mathbb{E}_{i \in \mathcal{T}}[F_{t_{N-r}+1}]} - \frac{1}{\mathbb{E}_{i \in \mathcal{T}}[F_{t_{N-r-1}+1}]}.$$

Following the same arguments, we can also show

$$\frac{1}{8dL_0R^2} \leq \frac{1}{\mathbb{E}_{i \in \mathcal{T}}[F_{t_{N-r-1}+1}]} - \frac{1}{\mathbb{E}_{i \in \mathcal{T}}[F_{t_{N-r-2}+1}]},$$

$$\ldots$$

$$\frac{1}{8dL_0R^2} \leq \frac{1}{\mathbb{E}_{i \in \mathcal{T}}[F_{t_1+1}]} - \frac{1}{\mathbb{E}_{i \in \mathcal{T}}[F_{t_1}]} \leq \frac{1}{\mathbb{E}_{i \in \mathcal{T}}[F_{t_1+1}]} - \frac{1}{F_0}.$$

Summing up all of them, we arrive at

$$\frac{N-r}{8dL_0R^2} \leq \frac{1}{\mathbb{E}_{i \in \mathcal{T}}[F_{t_{N-r}+1}]} - \frac{1}{F_0} \leq \frac{1}{\mathbb{E}_{i \in \mathcal{T}}[F_{t_{N-r}+1}]},$$

implying

$$\mathbb{E}_{i \in \mathcal{T}}[F_N] \leq \mathbb{E}_{i \in \mathcal{T}}[F_{t_{N-r}+1}] \leq \frac{8dL_0R^2}{N-r} \overset{r \leq N/2}{\leq} \frac{16dL_0R^2}{N}\mathbb{1}_{\{r \leq N/2\}}. \tag{49}$$

Combining equation 47 and equation 49 and taking the full expectation, we get

$$\mathbb{E}[F_N] \leq \left(1 - \frac{1}{4\sqrt{2}dL_1R}\right)^{N/2} F_0\mathbb{E}[\mathbb{1}_{\{r > N/2\}}] + \frac{16dL_0R^2}{N}\mathbb{E}[\mathbb{1}_{\{r \leq N/2\}}]$$

$$\leq \max\left\{\left(1 - \frac{1}{4\sqrt{2}dL_1R}\right)^{N/2} F_0, \frac{16dL_0R^2}{N}\right\},$$

which concludes the proof.

## D.2 PROOF OF THEOREM 4.3

Algorithm 5, presented in Section 4, uses the Golden Ratio Method (GRM) once per iteration. This method utilizes the oracle concept equation 5 (see Algorithm 6).

---

**Algorithm 6** Golden Ratio Method (GRM)

1: **Input:** interval $[a, b]$, accuracy $\hat{\epsilon}$
2: **Initialization:** define constant $\rho = \frac{1}{\Phi} = \frac{\sqrt{5}-1}{2}$
3: $y \leftarrow a + (1 - \rho)(b - a)$
4: $z \leftarrow a + \rho(b - a)$
5: **while** $b - a > \hat{\epsilon}$ **do**
6:    **if** $\phi(y, z) = -1$ **then**
7:       $b \leftarrow z$
8:       $z \leftarrow y$
9:       $y \leftarrow a + (1 - \rho)(b - a)$
10:    **else**
11:       $a \leftarrow y$
12:       $y \leftarrow z$
13:       $z \leftarrow a + \rho(b - a)$
14:    **end if**
15: **end while**
16: **Return:** $\frac{a+b}{2}$

---

We utilize the Golden Ratio Method to find a solution to the following one-dimensional problem (see line 2 in Algorithm 5):

$$\zeta_k = \arg\min_{\zeta \in \mathbb{R}} f(x^k + \zeta \mathbf{e}_{i_k}).$$

Using the well-known fact about the golden ratio method that GRM is required to do $N = \mathcal{O}\left(\log \frac{1}{\epsilon}\right)$ (where $\epsilon$ is the accuracy of the solution to the linear search problem in terms of the function value; due to equation 9, it is sufficient to take $\hat{\epsilon} = \frac{2\epsilon}{L_0}$), we derive the following corollaries from the solution of this problem: for simplicity, we consider the scenario when the golden ratio method solves the inner problem exactly ($\epsilon \simeq 0$). Then, we have the following:

$$f(x_k + \zeta_k \mathbf{e}_{i_k}) \leq f(x_k + \zeta \mathbf{e}_{i_k}), \qquad \forall \zeta \in \mathbb{R}. \tag{50}$$

Using the above inequality with $\zeta = \eta_k \nabla_{i_k} f(x^k)$, $\eta_k := \frac{1}{L_0 + L_1|\nabla_{i_k} f(x^k)|}$, and applying Assumption 1.4, we get

$$f(x^{k+1}) - f(x^k) = f(x^k + \zeta_k \mathbf{e}_{i_k}) - f(x^k)$$

$$\overset{equation\ 50}{\leq} f(x^k - \eta_k \nabla_{i_k} f(x^k)\mathbf{e}_{i_k}) - f(x^k)$$

$$\overset{equation\ 38}{\leq} -\frac{\eta_k}{2}\left(\nabla_{i_k} f(x^k)\right)^2. \tag{51}$$

The rest of the proof is identical to the proof given in Appendix D.1 and leads to the same bound:

$$\mathbb{E}[f(x^N)] - f^* \leq \max\left\{\left(1 - \frac{1}{4\sqrt{2}dL_1 R}\right)^{N/2} F_0, \frac{16dL_0 R^2}{N}\right\}.$$

The above upper bound implies that to achieve $\mathbb{E}[f(x^N)] - f^* \leq \varepsilon$, OrderRCD needs to perform

$$N = \mathcal{O}\left(\max\left\{\frac{dL_0 R^2}{\varepsilon}, dL_1 R \log \frac{F_0}{\varepsilon}\right\}\right) \qquad \text{iterations and}$$

$$T = \mathcal{O}\left(\max\left\{\frac{dL_0 R^2}{\varepsilon}, dL_1 R \log \frac{F_0}{\varepsilon}\right\} \cdot \log \frac{1}{\epsilon}\right) \qquad \text{Order Oracle calls,}$$

where $\epsilon$ is the accuracy of the solution of the auxiliary optimization problem (see line 2 in Algorithm 5), and it has to be sufficiently small.

## E  MISSING PROOF FOR GD IN THE STRONGLY CONVEX SETUP

From the analysis of the convex case, we have

$$f(x^{k+1}) - f(x^k) \overset{equation\ 14}{\leq} -\frac{1}{2(L_0 + L_1 \|\nabla f(x^k)\|)} \|\nabla f(x^k)\|^2. \tag{52}$$

Next, let us consider two cases: $\|\nabla f(x^k)\| \geq \frac{L_0}{L_1}$ and $\|\nabla f(x^k)\| < \frac{L_0}{L_1}$.

The case of $\|\nabla f(x^k)\| \geq \frac{L_0}{L_1}$. In this case, we have $L_0 + L_1 \|\nabla f(x^k)\| \leq 2L_1 \|\nabla f(x^k)\|$. Using this inequality in equation 52, we obtain

$$f(x^{k+1}) - f(x^k) \leq -\frac{\|\nabla f(x^k)\|}{4L_1}. \tag{53}$$

To continue the derivation, we also consider two possible situations depending on $F_k := f(x^k) - f^*$.

i) If $F_k \geq 1$, then we proceed as in the convex case and get

$$F_{k+1} \overset{equation\ 16}{\leq} \left(1 - \frac{1}{4L_1 R}\right) F_k. \tag{54}$$

In view of equation 52 and Lemma B.2, we have $F_{k+1} \leq F_k$ and $\|\nabla f(x^{k+1})\| \leq \|\nabla f(x^k)\|$. Therefore, if $F_k \geq 1$ and $\|\nabla f(x^k)\| \geq \frac{L_0}{L_1}$, then $F_t \geq 1$ and $\|\nabla f(x^t)\| \geq \frac{L_0}{L_1}$ for all $t = 0, 1, \ldots, k$. Let $\mathcal{T}_1$ be the largest $k \in [N-1]$ such that $F_k \geq 1$ and $\|\nabla f(x^k)\| \geq \frac{L_0}{L_1}$ (if there is no such $k$ for given initialization, then $\mathcal{T}_1 := -1$). Then, we have

$$F_{\mathcal{T}_1 + 1} \overset{equation\ 54}{\leq} \left(1 - \frac{1}{4L_1 R}\right)^{\mathcal{T}_1 + 1} F_0. \tag{55}$$

Using the above inequality, we can upper bound $\mathcal{T}_1$ as

$$\mathcal{T}_1 \leq 4L_1 R \log(F_0).$$

ii) If $F_k < 1$, we use Polyak-Łojasiewicz (Polyak, 1963; Łojasiewicz, 1963) inequality

$$\|\nabla f(x^k)\|^2 \geq 2\mu F_k \tag{56}$$

$$\overset{F_k < 1}{>} 2\mu (F_k)^2, \tag{57}$$

which follows from strong convexity Nesterov (2018). Then, we can continue the derivation as follows:

$$F_{k+1} \overset{equation\ 53}{\leq} F_k - \frac{1}{4L_1} \|\nabla f(x^k)\|$$

$$\overset{equation\ 57}{\leq} \left(1 - \frac{\sqrt{\mu}}{2\sqrt{2}L_1}\right) F_k. \tag{58}$$

Moreover, since equation 54 holds whenever $\|\nabla f(x^k)\| \geq \frac{L_0}{L_1}$, we can tighten the above inequality as

$$F_{k+1} \leq \left(1 - \max\left\{\frac{\sqrt{\mu}}{2\sqrt{2}L_1}, \frac{1}{4L_1 R}\right\}\right) F_k. \tag{59}$$

In view of Lemma B.2, we have $\|\nabla f(x^{k+1})\| \leq \|\nabla f(x^k)\|$. Therefore, if $\|\nabla f(x^k)\| \geq \frac{L_0}{L_1}$, then $\|\nabla f(x^t)\| \geq \frac{L_0}{L_1}$ for all $t = 0, 1, \ldots, k$. Let $\mathcal{T}_2$ be the largest $k \in [N-1]$ such that $\|\nabla f(x^k)\| \geq \frac{L_0}{L_1}$ (if there is no such $k$ for given initialization, then $\mathcal{T}_2 := -1$). Then, we have

$$F_{\mathcal{T}_2 + 1} \leq \left(1 - \max\left\{\frac{\sqrt{\mu}}{2\sqrt{2}L_1}, \frac{1}{4L_1 R}\right\}\right)^{\mathcal{T}_2 - \mathcal{T}_1} F_{\mathcal{T}_1 + 1}$$

$$\overset{equation\ 55}{\leq} \left(1 - \max\left\{\frac{\sqrt{\mu}}{2\sqrt{2}L_1}, \frac{1}{4L_1 R}\right\}\right)^{\mathcal{T}_2 - \mathcal{T}_1} \left(1 - \frac{1}{4L_1 R}\right)^{\mathcal{T}_1 + 1} F_0. \tag{60}$$

The case of $\left\|\nabla f(x^k)\right\| < \frac{L_0}{L_1}$. In this case, we have $L_0 + L_1 \|\nabla f(x^k)\| \leq 2L_0$. Using this inequality in equation 52, we obtain

$$F_{k+1} \overset{\text{equation 52}}{\leq} F_k - \frac{\left\|\nabla f(x^k)\right\|^2}{4L_0}$$

$$\overset{\text{equation 56}}{\leq} \left(1 - \frac{\mu}{2L_0}\right) F_k. \tag{61}$$

Since Algorithm 1 converges monotonically in terms of the gradient norm (see Appendix B), the above inequality holds for $k = \mathcal{T}_2 + 1, \mathcal{T}_2 + 2 \ldots, N - 1$ iterations and gives

$$F_N \leq \left(1 - \frac{\mu}{2L_0}\right)^{N-\mathcal{T}_2} F_{\mathcal{T}_2+1}$$

$$\overset{\text{equation 60}}{\leq} \left(1 - \frac{\mu}{2L_0}\right)^{N-\mathcal{T}_2} \left(1 - \max\left\{\frac{\sqrt{\mu}}{2\sqrt{2}L_1}, \frac{1}{4L_1 R}\right\}\right)^{\mathcal{T}_2-\mathcal{T}_1} \left(1 - \frac{1}{4L_1 R}\right)^{\mathcal{T}_1+1} F_0.$$

This concludes the proof.

## F   ADDITIONAL EXPLANATIONS REGARDING REMARK 1.3

We know from Remark 1.3 that there exist functions satisfying Assumption 1.2 when $L_0 = 0$. However, this problem does not reach a minimum (hence $R = \arg\inf f(x) = +\infty$). Therefore, in this section we show that, for example, gradient descent (Algorithm 1) will achieve the desired accuracy $\varepsilon$ in a finite number of iterations with exponential objective decrease.

We introduce the hyperparameter of the algorithm $R_s = \left\|x^0 - s\right\|$. Then we show exponential objective decrease to the desired accuracy by the example of gradient descent.

Using the Assumption 1.2 with $L_0 = 0$ we have:

$$f(x^{k+1}) - f(x^k) = f(x^k - \eta_k \nabla f(x^k)) - f(x^k)$$

$$\overset{\text{equation 8}}{\leq} -\eta_k \left\langle \nabla f(x^k), \nabla f(x^k) \right\rangle + \eta_k^2 \frac{L_1 \left\|\nabla f(x^k)\right\|}{2} \left\|\nabla f(x^k)\right\|^2$$

$$\overset{①}{\leq} -\eta_k \left\|\nabla f(x^k)\right\|^2 + \frac{\eta_k}{2} \left\|\nabla f(x^k)\right\|^2$$

$$= -\frac{\eta_k}{2} \left\|\nabla f(x^k)\right\|^2$$

$$= -\frac{1}{2L_1 \left\|\nabla f(x^k)\right\|} \left\|\nabla f(x^k)\right\|^2$$

$$= -\frac{1}{2L_1} \left\|\nabla f(x^k)\right\|, \tag{62}$$

where in ① we used $\eta_k \leq \frac{1}{L_1 \|\nabla f(x^k)\|}$.

Then using the convexity assumption of the function (see Assumption 1.5, $\mu = 0$), we have the following:

$$f(x^k) - f(s) \leq \left\langle \nabla f(x^k), x^k - s \right\rangle$$

$$\overset{\text{equation 7}}{\leq} \left\|\nabla f(x^k)\right\| \left\|x^k - s\right\|$$

$$\leq \left\|\nabla f(x^k)\right\| \underbrace{\left\|x^0 - s\right\|}_{R_s}.$$

Hence we have:

$$\left\|\nabla f(x^k)\right\| \geq \frac{f(x^k) - f(s)}{R_s}. \tag{63}$$

Then substituting equation 63 into equation 62 we obtain:

$$f(x^{k+1}) - f(x^k) \leq -\frac{1}{2L_1} \left\|\nabla f(x^k)\right\| \leq -\frac{1}{2L_1 R_s}(f(x^k) - f(s)).$$

This inequality is equivalent to the trailing inequality:

$$f(x^{k+1}) - f^* \le \left(1 - \frac{1}{2L_1 R_s}\right)(f(x^k) - f^*) + \frac{1}{2L_1 R_s}(f(s) - f^*). \tag{64}$$

Applying recursion to equation 64 we obtain:

$$f(x^N) - f^* \le \left(1 - \frac{1}{2L_1 R_s}\right)^N (f(x^0) - f^*) + f(s) - f^*.$$

Therefore, we have shown that Algorithm 1 will achieve the accuracy $f(s) - f^*$ in a finite number of iterations: $N = \mathcal{O}\left(L_1 R_s \log \frac{1}{\varepsilon}\right)$. $R_s$ is a finite number and increases as the desired accuracy improves. The same can be shown for other algorithms.