# OpenReview forum: "Exponential Objective Decrease in Convex Setup is Possible! Gradient Descent Method Variants under $(L_0,L_1)$-Smoothness"
_ICLR.cc/2026/Conference — Submitted to ICLR 2026_

### Official Review · Reviewer_5X4c · 2025-10-25

**Soundness:** 3
**Presentation:** 3
**Contribution:** 3
**Rating:** 4
**Confidence:** 3

**Summary:**

The paper studies gradient-based methods (GD, NGD, Clip-GD, RCD, and RCD with order oracle) for convex objectives under \((L_0,L_1)\)-smoothness. It establishes a clean two-phase behavior: when $\|\nabla f(x_k)\|\ge L_0/L_1$, the objective decreases exponentially; once below this threshold, the rate becomes sublinear. The story is extended from full-gradient methods to normalized/clipped variants and coordinate-oracle settings, with remarks for the strongly-convex case and small illustrative experiments.

**Strengths:**

- Clear regime characterization. With a stepsize like $\eta_k=(L_0+L_1\|\nabla f(x_k)\|)^{-1}$, the paper gives an interpretable ``fast-then-slow'' picture: exponential decrease above the $L_0/L_1$ threshold and standard $O(L_0R^2/N)$ behavior below it.

- Coverage across oracles. The analysis is propagated from full-gradient methods to NGD/Clip-GD and to RCD/OrderRCD, providing a unified treatment under $(L_0,L_1)$-smoothness.

- Tidy write-up. Assumptions and theorem statements are organized; examples help illustrate the phase transition.

**Weaknesses:**

- Topic saturation / mature techniques. Both the $L_0,L_1$-smoothness line and clipping/normalization ideas have been heavily studied; the contribution reads as a careful consolidation rather than a new mechanism.

- Incremental advance. The work largely systematizes known ingredients with refined constants and extensions across oracles; conceptual novelty is limited.

- Dominant sublinear regime near optimum. Even with an initial exponential phase, the practically dominant regime remains sublinear unless \(L_0\) is very small, tempering impact on broad convex problems.

- Dependence on thresholds/unknown constants. The results hinge on the $\|\nabla f(x_k)\|\ge L_0/L_1$ threshold and schedules that presume knowledge (or good estimates) of $L_0,L_1$; robustness when these are unknown is not fully addressed.

**Questions:**

I think the claimed geometric (linear) decrease in the $L_1$-dominated regime is not novel—it follows routinely from the standard $(L_0,L_1)$ descent calculus with gradient-normalized steps; if this assessment is mistaken, I welcome a correction.




In the $L_1$-dominated regime—either $L_0=0$ or whenever $\|\nabla f(x_k)\|\ge L_0/L_1$—the standard $(L_0,L_1)$ descent with the normalized stepsize $\eta_k=(L_0+L_1\|\nabla f(x_k)\|)^{-1}$ gives
$f(x_{k+1})-f^\star \le (1-\rho)[f(x_k)-f^\star]$ for some $\rho\in(0,1)$,
since $\frac{\|\nabla f\|^2}{L_0+L_1\|\nabla f\|}\ge c\,\|\nabla f\|$ once $\|\nabla f\|\ge L_0/L_1$.
Thus “geometric descent” above the threshold is a routine, well-known consequence of the $(L_0,L_1)$ calculus with gradient-normalized steps, and is not conceptually new; the only delicate part is the $L_0$-dominated small-gradient phase, where one typically recovers at best sublinear progress.

---

> ### Author Response · Authors · 2025-11-20
> **Respose to Reviewer 5X4c: Part 1**
>
> Dear **Reviewer 5X4c**,
>
> We thank you for your feedback on our work. We provide detailed answers to the comments and questions raised in the review below.
>
> >**Topic saturation / mature techniques. Both the $(L_0,L_1)$-smoothness line and clipping/normalization ideas have been heavily studied; the contribution reads as a careful consolidation rather than a new mechanism.**
>
> With all due respect, we disagree with this remark, particularly with the claim that the research direction on $(L_0,L_1)$-smoothness has been thoroughly studied. This area is relatively new (to the best of our knowledge, not even lower bounds for convex functions exist). Moreover, some competing works fail to explain the convergence behavior of algorithms in the regime where $|| \nabla f(x^k) || \geq L_0/L_1$, which is precisely what motivated our investigation.
>
> >**Incremental advance. The work largely systematizes known ingredients with refined constants and extensions across oracles; conceptual novelty is limited.**
>
> First and foremost, we would like to draw your attention to the fact that in practice, rapid convergence of algorithms is often observed in the initial iterations, followed by a slowdown to sublinear convergence. However, to the best of our knowledge, no theoretical explanation for this phenomenon existed prior to our work. For example, the optimal convergence rate (where the lower bound matches the upper bound, see [1]) for first-order algorithms on the class of $L$-smooth convex functions is sublinear. When considering the class of $(L_0,L_1)$-smooth convex functions, all known first-order algorithms also exhibit purely sublinear convergence (no summands with exponential objective decrease are present, see Table 1).
>
> Furthermore, our results imply that the phase of exponential objective decrease is notably pronounced in cases where $L_0/L_1$ is small or even zero (in the text, we note that there exist functions for which $L_0 = 0$).
>
> We also wish to note that RCD, OrderRCD, and standard stochastic algorithms share a common property—there are no guarantees of monotonic decrease in the gradient norm. Precisely for this reason, by building upon the ideas presented in the proofs for deterministic algorithms and the techniques developed for RCD and OrderRCD, it can be shown that a summands with  exponential objective decrease will also appear in the convergence bounds of stochastic algorithms.  Thus, we believe that the results presented in our work contain conceptual novelty, particularly as they can serve as a foundation for extending to stochastic problem settings, thereby improving the convergence results of standard stochastic algorithms.
>
> _[1] Nesterov, Y. (2018). Lectures on convex optimization (Vol. 137). Berlin: Springer International Publishing._
>
> >**Dominant sublinear regime near optimum. Even with an initial exponential phase, the practically dominant regime remains sublinear unless (L_0) is very small, tempering impact on broad convex problems.**
>
> As we mentioned previously, the theoretical justification for rapid function decay in the initial iterations is highly interesting. Furthermore, the regime identified for very small values of $L_0$ is particularly remarkable within the convex optimization framework.
>
> >**Dependence on thresholds/unknown constants. The results hinge on the $|| \nabla f(x_k) || \geq L_0/L_1$ threshold and schedules that presume knowledge (or good estimates) of $L_0,L_1$; robustness when these are unknown is not fully addressed.**
>
> The problem of estimating smoothness constants is not new (for instance, the same issue arises when estimating $L$ in standard smoothness). An example of analytical estimation of smoothness constants is provided in [2]. As an alternative, grid search can be employed. However, if we generalize ClipGD to the case with step size $\eta \leq (L_0 + L_1 c)^{-1}$, the arbitrary clipping radius $c$ eliminates the need to know the smoothness constants explicitly, though the convergence rate takes the following form: $f(x^N) - f^* \lesssim (1 - \frac{\eta c}{R})^T (f(x^0) - f^*) + \frac{R^2}{\eta (N-T)}$.
>
> _[2] Gorbunov, E., Tupitsa, N., Choudhury, S., Aliev, A., Richtárik, P., Horváth, S., & Takáč, M. (2024). Methods for convex $(l_0, l_1) $-smooth optimization: Clipping, acceleration, and adaptivity. arXiv preprint arXiv:2409.14989._

---

> > ### Author Response · Authors · 2025-11-20
> > **Respose to Reviewer 5X4c: Part 2**
> >
> > >**I think the claimed geometric (linear) decrease in the $L_1$-dominated regime is not novel—it follows routinely from the standard $(L_0,L_1)$ descent calculus with gradient-normalized steps; if this assessment is mistaken, I welcome a correction.**
> >
> > As we noted earlier, to the best of our knowledge, there were no results prior to our work that demonstrated summands with  exponential objective decrease in the convergence bounds of first-order algorithms for the convex regime.
> >
> > >**In the $L_1$-dominated regime—either $L_0 = 0$ or whenever $||\nabla f(x_k)|| \geq L_0/L_1$—the standard $(L_0, L_1)$ descent with the normalized stepsize $\eta_k = (L_0 + L_1 ||\nabla f(x_k)||)^{-1}$ gives $f(x_{k+1}) - f^* \leq (1 - \rho) [f(x_k) - f^*]$ for some $\rho \in (0,1)$, since $\frac{|| \nabla f||^2}{L_0 + L_1 || \nabla f||} \geq c, || \nabla f||$ once $|| \nabla f|| \geq L_0/L_1$. Thus “geometric descent” above the threshold is a routine, well-known consequence of the $(L_0,L_1)$ calculus with gradient-normalized steps, and is not conceptually new; the only delicate part is the $L_0$-dominated small-gradient phase, where one typically recovers at best sublinear progress.**
> >
> > You are correct in your reasoning, and this is precisely what we demonstrate in our work (please see the Appendix). We would like to reiterate that, to the best of our knowledge, our work is the first to substantiate these arguments, which is why we believe it contains conceptual novelty.
> >
> > With Respect,
> >
> > Authors

---

### Official Review · Reviewer_hV6d · 2025-10-30

**Soundness:** 3
**Presentation:** 3
**Contribution:** 2
**Rating:** 4
**Confidence:** 4

**Summary:**

The paper studies gradient descent (GD) and its variants (Normalized GD, Clipped GD, and Random Coordinate Descent) under the recently popularized $(L\_0, L\_1)$-smoothness assumption. The authors show that under this generalized smoothness, GD and its variants exhibit exponential objective decrease as long as the gradient norm is above a threshold ($\\|\nabla f(x)\\| \ge \frac{L\_0\}{L\_1}$), and revert to sublinear convergence afterwards.

**Strengths:**

- The result that GD can exhibit exponential objective decay in the convex setup under $(L\_0, L\_1)$-smoothness, particularly when $L\_0 = 0$, is conceptually interesting and highlights a nuanced behavior of gradient descent.

- The proofs and arguments are concise and technically sound.

- The exposition is organized and clear in most places, making it easy for the readers to follow the storyline of the paper.

**Weaknesses:**

1. *Significance of the results.* While the exponential objective value decay is mathematically neat, it only holds when the gradient norm is large. Once the iterates enter the small-gradient regime, the convergence becomes sublinear, which dominates asymptotically. In the end, the overall improvement is not substantial.
The extensions to Clipped GD and Random Coordinate Descent are provided, but they do not seem to constitute conceptually new insights beyond their analysis of full gradient descent.

2. *Lacking interpretation of the $L\_0 = 0$ case.*
The case of global linear convergence when $L\_0 = 0$ is emphasized as a key finding that distinguishes the result of this paper with similar existing works. But in that regime, the assumption $\\|\nabla f(x) - \nabla f(y)\\| \le L_1 \\|\nabla f(x)\\| \\|x - y\\|$ actually imposes a stronger constraint than standard Lipschitz smoothness near optima. Thus, the improvement stems from stronger assumptions rather than a fundamentally sharper analysis. I am not sure if such a setting is practically relevant.

**Questions:**

1. The standard $L$-smooth convex setting also satisfies $(L\_0, L\_1)$-smoothness if one sets $L\_0 = L$ and $L\_1 > 0$ arbitrarily. In that case, your results seem to suggest exponential decay in $f(x^k) - f^*$ with step size smaller than $1/L$, even though GD with step size less than $1/L$ is known to be sublinear (with a complexity lower bound). Could you clarify where this apparent contradiction is resolved? Can it be explained based on the structure of the worst case function for GD, or initialization?

2. I believe that a more proper way to state Theorem 3.1 is to first state a short preliminary lemma about monotonicity of $\\|\nabla f(x^k)\\|$ and then defining the transition point $T$ before or earlier within the theorem, rather than relegating them to the prose explanation following it. At the moment, it is not accurately stated. For example, the condition $\\|\nabla f(x^{N-1})\\| < \frac{L\_0}{L\_1}$ for the sublinear convergence part does not seem to reflect what the theorem actually tries to state. (Perhaps it was a typo for $\\|\nabla f(x^{0})\\| < \frac{L\_0}{L\_1}$.)

3. In Lines 79–86, $\lambda\_k$ is introduced without definition or context, making the introduction section non-self-contained for first time readers.

---

> ### Author Response · Authors · 2025-11-20
> **Respose to Reviewer hV6d: Part 1**
>
> Dear **Reviewer hV6d**,
>
> We thank you for your feedback on our work. We provide detailed answers to the comments and questions raised in the review below.
>
> >**Significance of the results. While the exponential objective value decay is mathematically neat, it only holds when the gradient norm is large. Once the iterates enter the small-gradient regime, the convergence becomes sublinear, which dominates asymptotically. In the end, the overall improvement is not substantial. The extensions to Clipped GD and Random Coordinate Descent are provided, but they do not seem to constitute conceptually new insights beyond their analysis of full gradient descent.**
>
> First and foremost, we would like to draw your attention to the fact that in practice, rapid convergence of algorithms is often observed in the initial iterations, followed by a slowdown to sublinear convergence. However, to the best of our knowledge, no theoretical explanation for this phenomenon existed prior to our work. For example, the optimal convergence rate (where the lower bound matches the upper bound, see [1]) for first-order algorithms on the class of $L$-smooth convex functions is sublinear. When considering the class of $(L_0,L_1)$-smooth convex functions, all known first-order algorithms also exhibit purely sublinear convergence (no summands with exponential objective decrease are present, see Table 1).
>
> Furthermore, our results imply that the phase of exponential objective decrease is notably pronounced in cases where $L_0/L_1$ is small or even zero (in the text, we note that there exist functions for which $L_0 = 0$).
>
> We also wish to note that RCD, OrderRCD, and standard stochastic algorithms share a common property—there are no guarantees of monotonic decrease in the gradient norm. Precisely for this reason, by building upon the ideas presented in the proofs for deterministic algorithms and the techniques developed for RCD and OrderRCD, it can be shown that a summands with  exponential objective decrease will also appear in the convergence bounds of stochastic algorithms. Thus, we believe that the algorithms presented in our work are significant, as they can serve as a foundation for extending to stochastic problem settings, thereby improving the convergence results of standard stochastic algorithms.
>
> _[1] Nesterov, Y. (2018). Lectures on convex optimization (Vol. 137). Berlin: Springer International Publishing._
>
> >**Lacking interpretation of the $L_0 = 0$ case. The case of global linear convergence when $L_0 =0 $ is emphasized as a key finding that distinguishes the result of this paper with similar existing works. But in that regime, the assumption $||\nabla f(x) - \nabla f(y)|| \leq L_1 ||\nabla f(x)|| ||x-y||$ actually imposes a stronger constraint than standard Lipschitz smoothness near optima. Thus, the improvement stems from stronger assumptions rather than a fundamentally sharper analysis. I am not sure if such a setting is practically relevant.**
>
> As we mentioned earlier, the possibility of linear convergence in the convex regime had not been explored prior to our work. We have identified a regime where this effect is observed. The key distinction from standard smoothness lies not in bounding the Hessian by a (globally pessimistic) constant, but in the fact that the local smoothness constant changes during iterations, thereby demonstrating adaptability in step size selection. Furthermore, it is important to note that functions satisfying this assumption do exist, which provides reasonable optimism for practical convergence improvements.
>
> >**The standard $L$-smooth convex setting also satisfies $(L_0,L_1)$-smoothness if one sets $L_0=L$ and $L_1>0$ arbitrarily. In that case, your results seem to suggest exponential decay in $f(x^k) - f^*$ with step size smaller than $1/L$, even though GD with step size less than $1/L$ is known to be sublinear (with a complexity lower bound). Could you clarify where this apparent contradiction is resolved? Can it be explained based on the structure of the worst case function for GD, or initialization?**
>
> The reviewer is right – if function is $L$-smooth, then it is $(L, 0)$-smooth and, therefore, it is also $(L, L_1)$-smooth with arbitrary $L_1 > 0$. However, our result holds for GD with stepsize $\sim (L_0 + L_1 || \nabla f(x^k) ||)$, while the existing worst-case guarantees for GD do not hold for such stepsizes. Moreover, the existing convergence guarantees for GD under $L$-smoothness and convexity do not take into account the size of the initial gradient norm, which is important for showing exponential convergence during the initial stage.

---

> > ### Author Response · Authors · 2025-11-20
> > **Respose to Reviewer hV6d: Part 2**
> >
> > >**I believe that a more proper way to state Theorem 3.1 is to first state a short preliminary lemma about monotonicity of $||\nabla f(x^k)||$ and then defining the transition point $T$ before or earlier within the theorem, rather than relegating them to the prose explanation following it. At the moment, it is not accurately stated. For example, the condition $||\nabla f(x^{N-1})|| \leq \frac{L_0}{L_1}$ for the sublinear convergence part does not seem to reflect what the theorem actually tries to state. (Perhaps it was a typo for $||\nabla f(x^{0})|| \leq \frac{L_0}{L_1}$.)**
> >
> > We would like to thank you for this remark. Indeed, adding to the theorem text the presence of monotonic decrease in the gradient norm, as well as introducing
> > $T = \min \left\[ k \in \{0,1,\dots, N-1\} \quad | \quad || \nabla f(x^k) || < L_0/L_1 \right\]$ in the  exponential objective decrease block
> >
> > $$ f(x^T) - f^* \leq \left( 1 - \frac{1}{4L_1 R} \right)^T F_0 $$
> >
> > and clarifying the sublinear convergence block for the case when $|| \nabla f(x^k) || < L_0 / L_1$:
> > $$
> > f(x^{N}) - f^* < \frac{4 L_0 R^2}{N-T}
> > $$
> > will improve the presentation of our results.
> >
> > >**In Lines 79–86, $\lambda_k$ is introduced without definition or context, making the introduction section non-self-contained for first time readers.**
> >
> > We agree that explicitly defining the clipping operator before lines 79–86 will enhance the self-containedness and presentation of the paper.
> >
> > With Respect,
> >
> > Authors

---

### Official Review · Reviewer_zerM · 2025-10-30

**Soundness:** 3
**Presentation:** 3
**Contribution:** 2
**Rating:** 4
**Confidence:** 3

**Summary:**

This paper studies gradient-based optimization methods (GD, NGD, Clip-GD, RCD, and OrderRCD) under the (L₀, L₁)-smoothness assumption, a generalization of Lipschitz smoothness where the smoothness constant depends linearly on the gradient norm. The key result is that exponential (linear) convergence rates can occur in convex problems when the gradient norm satisfies $\|f(x_k)\|\geq L_0/L_1$ even without assuming strong convexity.
The paper systematically analyzes GD, normalized GD, clipped GD, random coordinate descent, OrderRCD, and extend their results to the strongly convex case. For GD they show that their method achieves an exponential decrease when above the threshold. For normalized GD they show a similar rate. For random coordinate descent and OrderRCD they give the first analysis for these families of functions. They support their theoretical results with experiments.

**Strengths:**

1. The discovery and proof that exponential convergence can arise in convex settings under generalized smoothness is both non-trivial and potentially impactful.
2. Provides first convergence analyses under this generalized smoothness assumption, for RCD and OrderRCD
3. Shows how to get exponential rates of convergence for non strongly convex functions.

**Weaknesses:**

1. While rigorous, the results could use more intuitive explanations of why (L₀, L₁)-smoothness leads to exponential decrease.
2. If the gradient norm is large, isn’t gradient descent always going to make a large progress? I am not sure why this is new.
3. When was this kind of smoothness introduced? Based on the related works it seems to be quite recent.
4. Are these methods useful in practice over SGD in large data sets?
5. How would one estimate L_0 and L_1 in practice as they are needed in the step sizes?

**Questions:**

See weaknesses!

---

> ### Author Response · Authors · 2025-11-20
> **Respose to Reviewer zerM**
>
> Dear **Reviewer zerM**,
>
> We thank you for your feedback on our work. We provide detailed answers to the comments and questions raised in the review below.
>
> >**While rigorous, the results could use more intuitive explanations of why (L₀, L₁)-smoothness leads to exponential decrease.**
>
> We thank the reviewer for this insightful suggestion. We agree that providing more intuitive explanations would enhance the paper's accessibility. Below we outline the key intuition behind why $(L_0, L_1)$-smoothness enables exponential decrease.
>
> The $(L_0, L_1)$-smoothness condition
> can be interpreted as allowing the local smoothness to scale with the current gradient norm. This creates a self-amplifying effect: when $|| \nabla f(x)||$ is large, the effective smoothness constant $L_0 + L_1 ||\nabla f(x)||$ is large. This permits larger step sizes while maintaining stability, leading to rapid progress; The rapid progress further reduces $||\nabla f(x)||$, creating a virtuous cycle of acceleration.
>
> >**If the gradient norm is large, isn’t gradient descent always going to make a large progress? I am not sure why this is new.**
>
> We would like to draw your attention to the fact that in practice, rapid convergence of algorithms is often observed in the initial iterations, followed by a slowdown to sublinear convergence. However, to the best of our knowledge, no theoretical explanation for this phenomenon existed prior to our work. For example, the optimal convergence rate (where the lower bound matches the upper bound, see [1]) for first-order algorithms on the class of $L$-smooth convex functions is sublinear. When considering the class of $(L_0,L_1)$-smooth convex functions, all known first-order algorithms also exhibit purely sublinear convergence (no summands with  exponential objective decrease are present, see Table 1).
>
> _[1] Nesterov, Y. (2018). Lectures on convex optimization (Vol. 137). Berlin: Springer International Publishing._
>
> >**When was this kind of smoothness introduced? Based on the related works it seems to be quite recent.**
>
> Yes, the research direction related to $(L_0,L_1)$-smoothness is relatively new. The first work that introduced this assumption is [2]. Zhang et al. observed that local smoothness constants correlate with the gradient norm along the training trajectory for AWD-LSTM.
>
> _[2] Zhang, J., He, T., Sra, S., & Jadbabaie, A. (2019). Why gradient clipping accelerates training: A theoretical justification for adaptivity. arXiv preprint arXiv:1905.11881._
>
> >**Are these methods useful in practice over SGD in large data sets?**
>
> It is worth noting that the algorithms discussed in Section 3 are deterministic. In Section 4, we considered a class of coordinate algorithms that can be classified as stochastic due to the randomness in selecting the active coordinate. We wish to emphasize that RCD, OrderRCD, and SGD share a common property—there are no guarantees of monotonic decrease in the gradient norm. Precisely for this reason, by building upon the idea presented in the proof for deterministic algorithms and the technique used in the proofs for RCD and OrderRCD, it can be shown that a summands with  exponential objective decrease will also appear in the convergence bounds of stochastic algorithms. Thus, we believe that the algorithms presented in our work are valuable, as they can serve as a foundation for extending to the stochastic problem setting, thereby improving convergence results for SGD on large datasets.
>
> >**How would one estimate L_0 and L_1 in practice as they are needed in the step sizes?**
>
> The problem of estimating smoothness constants is not new (for instance, the same issue arises when estimating $L$ in standard smoothness). An example of analytical estimation of smoothness constants is provided in [3]. As an alternative, grid search can be employed. However, if we generalize ClipGD to the case with step size $\eta \leq (L_0 + L_1 c)^{-1}$, the arbitrary clipping radius $c$ eliminates the need to know the smoothness constants explicitly, though the convergence rate takes the following form: $f(x^N) - f^* \lesssim (1 - \frac{\eta c}{R})^T (f(x^0) - f^*) + \frac{R^2}{\eta (N-T)}$.
>
> _[3] Gorbunov, E., Tupitsa, N., Choudhury, S., Aliev, A., Richtárik, P., Horváth, S., & Takáč, M. (2024). Methods for convex $(l_0, l_1) $-smooth optimization: Clipping, acceleration, and adaptivity. arXiv preprint arXiv:2409.14989._
>
>
> With Respect,
>
> Authors

---

### Official Review · Reviewer_ZcXk · 2025-11-01

**Soundness:** 3
**Presentation:** 3
**Contribution:** 3
**Rating:** 6
**Confidence:** 3

**Summary:**

The paper analyzes (unconstrained) convex optimization under the recently popularized $(L_0,L_1)$-smoothness assumption. For convex functions, the paper demonstrates a two-phase behavior for simple first-order methods. In particular, when the gradient norm is above $L_0/L_1$, gradient descent can achieve exponential (linear) decrease of the objective even for the (non-strongly) convex setup. Below that threshold, the rate reverts to the usual sublinear $O(1/N)$ regime. The same phenomenon is established for other first-order methods of normalized and clipped gradient descent, refining previous analysis on $(L_0,L_1)$-smooth functions. The paper additionally shows similar results for random coordinate descent and a sign-oracle variant (OrderRCD) which is new for $(L_0,L_1)$-smooth functions, includes results for strongly convex functions (which seems like a three-phase linear convergence rate) and a simple numerical experiment.

**Strengths:**

- The work extensively covers various types of first-order algorithms including classical methods, refined methods designed for or closely related to $(L_0,L_1)$-smooth functions (clipGD, etc.) and coordinate descent methods.
- Comparison between the new results and previous work seem to be clearly presented.

**Weaknesses:**

See **Questions.**

**Questions:**

- As the main result of this paper is the discovery of the two-phase analysis of first-order methods, can the authors explain a bit more on what would be the takeaway or benefits of having linear convergence in a region far from the optimum?
    - It’s of course nice to know more details about the convergence dynamics and I am aware that the paper’s focus is on the theory side, but I can’t see how this can be an “extremely” interesting result yet (compared to previous $(L_0, L_1)$-smooth convergence results). I also guess that we won’t be able to see similar phenomena for more realistic nonconvex functions (or if so, please tell me).
    - The two-phase analysis might turn out to be something interesting if this theory either better explains the actual dynamics instantiated by first-order algorithms for general $(L_0, L_1)$-smooth functions other than the simple example in Section 7, or motivates ways to accelerate in the linear convergence regime to enter the sublinear neighborhood faster, both of which I think are possible to check. Can the authors comment on these aspects?
- For coordinate descent, is it correct that we cannot expect to extend these to general stochastic gradient methods? While extending to stochastic first-order methods could be an interesting future direction, but the proof technique seems to be specific to coordinate descent methods. (I don’t really consider this as a weakness, I am just wondering if there is a possibility to manipulate terms so that we get a similar descent inequality with additional noise terms for stochastic oracles, but it seems hard and I wanted to check if this guess is right.)
- Minor: What is $N_3$ in Theorem 5.1? I might have missed something, but I can’t see how this is used in the theorem statement.

---

> ### Author Response · Authors · 2025-11-20
> **Respose to Reviewer ZcXk**
>
> Dear **Reviewer ZcXk**,
>
> We thank you for your interest in our work. We attach below detailed responses to the questions raised in the review.
>
> >**As the main result of this paper is the discovery of the two-phase analysis of first-order methods, can the authors explain a bit more on what would be the takeaway or benefits of having linear convergence in a region far from the optimum?**
>
> First and foremost, we would like to draw your attention to the fact that in practice, rapid convergence of algorithms is often observed in the initial iterations, followed by a slowdown to sublinear convergence. However, to the best of our knowledge, no theoretical explanation for this phenomenon existed prior to our work. For example, the optimal convergence rate (where the lower bound matches the upper bound, see [1]) for first-order algorithms on the class of $L$-smooth convex functions is sublinear. When considering the class of $(L_0,L_1)$-smooth convex functions, all known first-order algorithms also exhibit purely sublinear convergence (no summands with exponential objective decrease are present, see Table 1).
>
> Furthermore, our results imply that the phase of exponential objective decrease is notably pronounced in cases where $L_0/L_1$ is small or even zero (in the text, we note that there exist functions for which $L_0 = 0$).
>
> _[1] Nesterov, Y. (2018). Lectures on convex optimization (Vol. 137). Berlin: Springer International Publishing._
>
> >**It’s of course nice to know more details about the convergence dynamics and I am aware that the paper’s focus is on the theory side, but I can’t see how this can be an “extremely” interesting result yet (compared to previous $(L_0,L_1)$-smooth convergence results). I also guess that we won’t be able to see similar phenomena for more realistic nonconvex functions (or if so, please tell me).**
>
> **On Theoretical Optimality Criteria of Algorithms**: Due to the inclusion of summands with  exponential objective decrease, we improve the iteration complexity of GD, NGD, and ClipGD compared to known results for $(L_0,L_1)$-smooth functions. Furthermore, we are the first to derive convergence bounds for a coordinate algorithm and for OrderRCD, where the latter algorithm operates under a highly restricted (comparative) oracle used in Human Feedback applications (e.g., RLHF).
>
> **Regarding Numerical Experiments**: In Figure 2, we demonstrated the importance of step-size selection and the possibility of achieving exponential objective decrease up to a desired accuracy.
>
> **In the Non-Convex Setting**: Two convergence regimes (both sublinear) are also observed. For example, for GD with step size $\eta_k = (L_0 + L_1 |\nabla f(x^k)|)^{-1}$, the following convergence rate is obtained:$\frac{1}{N} \sum_{k=1}^{N} || \nabla f(x^k) || \lesssim \frac{4 L_1 F_0}{N} + \sqrt{\frac{4L_0 F_0}{N}}$, where $F_0 = f(x^0) - f^*$.
>
> >**The two-phase analysis might turn out to be something interesting if this theory either better explains the actual dynamics instantiated by first-order algorithms for general $(L_0,L_1)$-smooth functions other than the simple example in Section 7, or motivates ways to accelerate in the linear convergence regime to enter the sublinear neighborhood faster, both of which I think are possible to check. Can the authors comment on these aspects?**
>
> In our view, we believe we have already addressed this question (please refer to our response to the first question).
>
> >**For coordinate descent, is it correct that we cannot expect to extend these to general stochastic gradient methods? While extending to stochastic first-order methods could be an interesting future direction, but the proof technique seems to be specific to coordinate descent methods. (I don’t really consider this as a weakness, I am just wondering if there is a possibility to manipulate terms so that we get a similar descent inequality with additional noise terms for stochastic oracles, but it seems hard and I wanted to check if this guess is right.)**
>
> The coordinate descent-type algorithms considered can be classified as stochastic due to the randomness in selecting the active coordinate. It is worth noting that general stochastic gradient methods and the algorithms discussed in Section 4 share a common characteristic: due to their stochastic nature, there are no guarantees of gradient monotonicity. We believe that based on the technique presented in our work (which enables the unification of the two convergence regimes), it is possible to obtain similar convergence results for general stochastic gradient methods.
>
> >**Minor: What is $N_3$ in Theorem 5.1? I might have missed something, but I can’t see how this is used in the theorem statement.**
>
> Yes, thank you. $N_3$ is neither used nor defined anywhere. We will remove it from Theorem 5.1.
>
>
> With Respect,
>
> Authors

---

### Author Response · Authors · 2025-11-26

Dear **Reviewers**,

With this message, we would like to kindly remind you about our responses submitted one week ago. We hope that we have adequately addressed the questions raised in your reviews. If anything remains unclear or requires further clarification, please let us know. We will be happy to provide any additional explanations you may need.

With Respect,

Authors

---

### Meta-Review · Area_Chair_AvDm · 2026-01-08

**Summary:**

All reviewers agree that the paper provides a technically improved convergence analysis of gradient descent and several of its variants (NGD, Clip-GD, RCD, and OrderRCD) under the recently introduced $(L_0, L_1)$-smoothness assumption.

The main point of concern among reviewers is the **significance and novelty** of the results, which were not adequately addressed in the rebuttal. For instance, Reviewer ZcXk asked about the takeaway or benefits of having linear convergence in a region far from the optimum. the authors mainly emphasized that no theoretical explanation for this phenomenon existed prior to their work, but did not provide further discussion on its practical implications or applications. In addition, Reviewers ZcXk, zerM, and hV6d all raised concerns about extending the analysis to stochastic problem settings. These concerns were not sufficiently addressed: in particular, the rebuttal does not provide a solid argument supporting the authors’ claim that their method can serve as a foundation for extending to stochastic problem settings, thereby improving the convergence results of standard stochastic algorithms.

**Reviewer Concerns:**

**Reviewer ZcXk’s concerns:**

(1) The concern regarding the takeaway or benefits of having linear convergence in a region far from the optimum was partially addressed. In particular, the authors mainly emphasized that no theoretical explanation for this phenomenon existed prior to their work, but did not provide further discussion on its practical implications or applications.

(2) The concern regarding a clearer explanation of the two-phase analysis, supported by additional experiments beyond the simple illustrative example in Section 7, was not addressed.

(3) The concern regarding the extension to stochastic problem settings was only partially addressed. In particular, there is no solid argument supporting the authors’ claim that their method can serve as a foundation for extending to stochastic problem settings, thereby improving the convergence results of standard stochastic algorithms.

(4) The concern regarding $N_3$ in Thm. 5.1 was addressed.

**Reviewer zerM’s concerns:**

(1) The concern regarding the intuitive explanation for the exponential decrease was not addressed. In particular, the authors’ explanation: "This creates a self-amplifying effect: when $|| \nabla f(x) ||$ is large, the effective smoothness constant $L_0+L_1 || \nabla f(x) ||$ is large. This permits larger step sizes while maintaining stability, leading to rapid progress" is inconsistent with their own stepsize rule in Theorem 3.1. Indeed, Theorem 3.1 uses $\eta_k=(L_0+L_1|| \nabla f(x) ||)^{-1}$, which decreases as $|| \nabla f(x) ||$ increases. Therefore, the claimed “larger step sizes when the gradient is large” mechanism does not apply, and a corrected intuitive explanation (or a revised statement) is needed.

(2) The concern regarding gradient descent was partially addressed by explaining the comparison of convergence rates for Clip-GD in the convex case, as summarized in Table 1.

(3) The concern regarding the history of the $(L_0, L_1)$-smoothness condition was addressed through additional clarification provided by the authors.

(4) The concern regarding the extension to stochastic problem settings was only partially addressed. In particular, there is no solid argument supporting the authors’ claim that their method can serve as a foundation for extending to stochastic problem settings, thereby improving the convergence results of standard stochastic algorithms.

(5) The concern regarding the dependence on the estimation of $L_0$ and $L_1$ was not addressed. Specifically, the upper bound on the stepsize proposed in the response still depends explicitly on the unknown constants $L_0$ and $L_1$.

**Reviewer hV6d’s concerns:**

(1) The concern regarding the significance of the results was only partially addressed, as there is no solid argument supporting the authors’ claim that their method can serve as a foundation for extending to stochastic problem settings, thereby improving the convergence results of standard stochastic algorithms.

(2) The concern regarding the lack of interpretation of the case $L_0=0$ was not adequately addressed. The authors merely stated that functions satisfying this assumption exist, without further explanation. However, if $L_0=0$ in the $(L_0, L_1)$-smoothness condition, then there is no point $x^\ast$ such that $\nabla f(x^\ast)=0$, unless $f$ is a constant function.

(3) The concern regarding the standard $L$-smooth convex setting was addressed.

(4) The concerns regarding the statement of Theorem 3.1 and the notation $\lambda_k$ were addressed through additional clarification.

**Reviewer 5X4c’s concerns:**

(1) The concern regarding novelty was only partially addressed. In particular, there is no solid argument supporting the authors’ claim that their method can serve as a foundation for extending to stochastic problem settings, thereby improving the convergence results of standard stochastic algorithms. Moreover, for the case $L_0=0$, the authors merely stated that functions satisfying this assumption exist, without further explanation. However, under the $(L_0, L_1)$-smoothness condition, if $L_0=0$, then there is no point $x^\ast$ such that $\nabla f(x^\ast)=0$, unless $f$ is a constant function.

(2) The concern regarding the dependence on thresholds or unknown constants was not addressed. Specifically, the upper bound on the step size provided in the response still depends on the unknown constants $L_0,L_1$.

**Reviewer Scores:**

**Reviewer ZcXk** may maintain a score of 6 or decrease the score (likely from 6 to 4), as the main concerns raised in the review were only partially addressed in the rebuttal.

**Reviewer zerM**, **Reviewer hV6d** and **Reviewer 5X4c** would likely maintain their scores of 4, since several of their concerns were not adequately addressed in the rebuttal.

---

### Decision · Program_Chairs · 2026-01-26

Reject